# Fast Bayesian Estimation of Point Process Intensity as Function of Covariates

**Hideaki Kim**      **Taichi Asami**      **Hiroyuki Toda**[*]
NTT Human Informatics Laboratories
NTT Corporation
{`hideaki.kin.cn, taichi.asami.ka`}`@hco.ntt.co.jp, hirotoda@acm.org`

## Abstract

In this paper, we tackle the Bayesian estimation of point process intensity as a function of covariates. We propose a novel augmentation of permanental process called *augmented permanental process*, a doubly-stochastic point process that uses a Gaussian process on covariate space to describe the Bayesian a priori uncertainty present in the square root of intensity, and derive a fast Bayesian estimation algorithm that scales linearly with data size without relying on either domain discretization or Markov Chain Monte Carlo computation. The proposed algorithm is based on a non-trivial finding that the representer theorem, one of the most desirable mathematical property for machine learning problems, holds for the augmented permanental process, which provides us with many significant computational advantages. We evaluate our algorithm on synthetic and real-world data, and show that it outperforms state-of-the-art methods in terms of predictive accuracy while being substantially faster than a conventional Bayesian method.

## 1    Introduction

Point processes are widely-used statistical tools to analyze point patterns observed on temporal, spatial, and spatio-temporal domains. They have a wide ranging list of applications in neuroscience [9], finance [6], epidemiology [14], ecology [41], and others.

The essential task in point processes is to estimate from event data an *intensity* function, that is, an instantaneous probability of events occurring at each point in the observation domain. Generally, there are two kinds of scenarios in the estimation task: one is to estimate intensities as functions of every point in the observation domain; the other is to estimate intensities as functions of covariates with the assumption that covariates are given at every point in the observation domain. For example, in ecological applications where the event points are the locations of trees, the former estimates the value of intensity at each geographical point (longitude and latitude), while the latter estimates the intensity as function of spatial covariates such as slope, elevation, and humidity, each of which varies across the geographical domain. The latter scenario has greater practical importance because it enables us to investigate factors that may be causing events. But unfortunately, it has been overlooked in the recent machine learning community. This paper focuses on using the function of covariates to solve the estimation problem of intensity.

The simplest and most classical model used to estimate intensities as functions of covariates is the Cox proportional hazards model [8, 43, 10] (or the log-linear model), which forms the logarithm of the intensity function as a linear combination of covariates. Although this parametric model is very easy to handle, it fails to address the nonlinear dependence of point patterns on covariates, which is often the case in real-world data. To rectify this limitation, two nonparametric approaches have been

36th Conference on Neural Information Processing Systems (NeurIPS 2022).

---

[*]Current affiliation is Yokohama City University.

proposed: Kernel-based intensity estimator (KIE) [17, 18, 5], which is a rescaled version of kernel density estimator, is the gold standard in estimating intensity as a function of covariates in a non-Bayesian way; it is easy to implement and rapid in computation. A Bayesian alternative is Gaussian Cox process (GCP) [29, 37, 22, 21, 28, 51], in which a flexible prior over a latent intensity is established by a Gaussian process/Gaussian random field [35] through a positive link function. GCPs are promising approaches as they allow the estimation of the uncertainty of intensity and the hyper-parameter optimization in a statistically formal way. However, the Bayesian estimation for GCPs is highly challenging as the likelihood function depends on the functional form of the latent intensity over a compact domain, which imposes the intractable computation of infinite-dimensional distributions. Most algorithms published to date have dealt with the difficulty by using domain discretization or/and Markov Chain Monte Carlo (MCMC). But domain discretization suffers from poor scaling in terms of domain dimension, as well as being sensitive to discretization size. Moreover, MCMC has significant computation demands, making GCPs less attractive than KIEs.

In this paper, for the problem of estimating intensity as function of covariates, we propose a novel Bayesian estimation scheme for GCPs with quadratic link functions without recourse to either domain discretization or MCMC. GCPs with quadratic link functions are called *permanental processes* [27] when the Gaussian process for intensity is defined in the observation domain. In our proposed model, the Gaussian process for intensity should be defined in the covariate domain, and thus we call it the *augmented permanental process* (APP). We show that the representer theorem [46], one of the most desirable mathematical property for machine learning problems, holds for the APP, which is one of the main contributions of this paper. The finding provides us with many significant computational advantages, resulting in a fast Bayesian estimation scheme that scales linearly with the number of data points. Furthermore, based on a Laplace approximation in the functional space, we derive the predictive distribution and the marginal likelihood in a feasible form, which enables us to implement the uncertainty evaluation and the hyper-parameter optimization in a Bayesian manner.

In Section 2, we introduce APP and construct a scalable Bayesian estimation scheme for it. In Section 3, we outline related work. In Section 4, we compare APP against KIE- and GCP-based algorithms on synthetic and real-world data, and confirm that our scheme achieves higher predictive accuracy than the state-of-the-art algorithms as well as substantial speed improvements over the conventional GCP-based algorithm. Finally, Section 5 states our conclusions.

## 2 Methods

### 2.1 Augmented Permanental Processes

Suppose that a dataset of $N$ point events and a map of $D_y$-dimensional covariate, denoted by $\mathcal{D} = \{\boldsymbol{t}_n \in \mathcal{T}\}_{n=1}^N$ and $\boldsymbol{y}(\boldsymbol{t}) : \mathcal{T} \to \mathcal{Y} \subset \mathbb{R}^{D_y}$, respectively, are observed for a compact $D_t$-dimensional domain, $\mathcal{T} \subset \mathbb{R}^{D_t}$. We consider a variant of Gaussian Cox processes which assumes that latent function, $x(\boldsymbol{y}) : \mathcal{Y} \to \mathbb{R}$, is generated from a Gaussian process (GP) on $\mathcal{Y}$, and point events are generated on $\mathcal{T}$ from a point process with *intensity* function, $\lambda(\boldsymbol{t})$, so that

$$p(x(\boldsymbol{y})|\mathcal{D}) = \frac{p(\mathcal{D}|x(\boldsymbol{y}))\,\mathcal{GP}(x(\boldsymbol{y})|k)}{\int \mathscr{D}x(\boldsymbol{y})\,p(\mathcal{D}|x(\boldsymbol{y}))\,\mathcal{GP}(x(\boldsymbol{y})|k)}, \tag{1}$$

$$\log p(\mathcal{D}|x(\boldsymbol{y})) = \sum_{n=1}^N \log \lambda(\boldsymbol{t}_n) - \int_{\mathcal{T}} \lambda(\boldsymbol{t})d\boldsymbol{t}, \quad \lambda(\boldsymbol{t}) = \kappa\big(x(\boldsymbol{y}(\boldsymbol{t}))\big), \tag{2}$$

where $\mathcal{GP}(x(\boldsymbol{y})|k)$ represents a GP with kernel function $k(\boldsymbol{y}, \boldsymbol{y}')$, $p(\mathcal{D}|x(\boldsymbol{y}))$ is the likelihood function of the point process, $\kappa(x) : \mathbb{R} \to \mathbb{R}_+$ is a non-negative function called the *link function*, while $\int \mathscr{D}x(\boldsymbol{y})$ in the denominator represents the integral over the function or the infinite-dimensional variable $x(\boldsymbol{y})$. Among popular link functions such as exponential and sigmoidal, this paper adopts the quadratic form of the link function, $\kappa(x) = x^2$, for its computing benefits. We call the model defined by (1-2) with the quadratic link function the *augmented permanental process* (APP), because it reduces to the ordinary permanental process when the covariate map is the identity mapping $\boldsymbol{y}(\boldsymbol{t}) = \boldsymbol{t}$. Intensity $\lambda(\boldsymbol{t})$ represents an instantaneous probability of events occurring at each point on $\mathcal{T}$, and our goal is to estimate, based on APP, the functional form of the intensity over covariate domain, $\kappa(x(\boldsymbol{y})) = x^2(\boldsymbol{y})$, from the observed data, $\{\mathcal{T}, \{\boldsymbol{t}_n\}_{n=1}^N, \boldsymbol{y}(\boldsymbol{t})\}$. For illustration, see Figure S3 in the supplementary material.

In general, Bayesian estimation on normal Gaussian Cox processes, equivalent to (1-2) with $\boldsymbol{y}(\boldsymbol{t}) = \boldsymbol{t}$, is often referred as "*doubly-intractable*", because it requires solving the integral of the stochastic function $x(\boldsymbol{y})$ as well as computing the intractable posterior probability of an infinite-dimensional variable. Furthermore, APP is challenged by one more difficulty: the domain on which the GP is defined ($\mathcal{Y}$) differs from that over which the intensity integral is performed ($\mathcal{T}$), which degrades mathematical tractability. Below, we address this "*triply-intractable*" problem with the help of *path integral* representation, a mathematical tool that has been proposed most recently [24].

## 2.2 Maximum *A Posteriori* Estimator

We consider the problem of obtaining the maximum *a posteriori* (MAP) estimator of $x(\boldsymbol{y})$ that maximizes the posterior probability (1). First, we solve the discrepancy between the domains for the GP in (1) and the intensity integral in the likelihood function (2), as it makes functional analyses problematic. If the covariate map $\boldsymbol{y}(\boldsymbol{t})$ is bijective, it is trivial that the intensity integral over $\mathcal{T}$ in (2) can be transformed into the integral over the covariate domain as, $\int_{\mathcal{T}} \cdot d\boldsymbol{t} = \int_{\mathcal{Y}} \cdot |\boldsymbol{J_y}^{-1}| d\boldsymbol{y}$, where $\boldsymbol{J_y}$ is the Jacobian matrix of the map $\boldsymbol{y}(\boldsymbol{t})$. We can generalize the transformation for a non-bijective map, which often happens in practice (e.g. the value of a covariate varies periodically over time), as follows:

$$\int_{\mathcal{T}} \lambda(\boldsymbol{t}) d\boldsymbol{t} = \int_{\mathcal{T}} x^2(\boldsymbol{y}(\boldsymbol{t})) d\boldsymbol{t} = \int_{\mathcal{T}} d\boldsymbol{t} \int_{\mathcal{Y}} x^2(\boldsymbol{y}) \delta(\boldsymbol{y}(\boldsymbol{t}) - \boldsymbol{y}) d\boldsymbol{y} = \int_{\mathcal{Y}} \rho(\boldsymbol{y}) x^2(\boldsymbol{y}) d\boldsymbol{y}, \quad (3)$$

where $\delta(\cdot)$ is the Dirac delta function, and $\rho(\boldsymbol{y}) = \int_{\mathcal{T}} \delta(\boldsymbol{y}(\boldsymbol{t}) - \boldsymbol{y}) d\boldsymbol{t}$. We call $\rho(\boldsymbol{y})$ the *weight distribution of the covariate*. Actually, our proposed algorithm does not need to evaluate $\rho(\boldsymbol{y})$ directly, and thus we do not go into detail about how to compute $\rho(\boldsymbol{y})$. Rather, we emphasize that the following transformation holds for arbitrary functions of covariates, $f(\boldsymbol{y})$:

$$\int_{\mathcal{Y}} \rho(\boldsymbol{y}) f(\boldsymbol{y}) d\boldsymbol{y} = \int_{\mathcal{T}} f(\boldsymbol{y}(\boldsymbol{t})) d\boldsymbol{t}. \quad (4)$$

Next, we derive the MAP estimator of $x(\boldsymbol{y})$, denoted by $\hat{x}(\boldsymbol{y})$, by using the path integral representation of GP [24],

$$\mathcal{GP}(x(\boldsymbol{y})|k) \, \mathscr{D}x(\boldsymbol{y}) = \sqrt{\frac{1}{|\mathcal{K}|}} \exp\left[-\frac{1}{2} \iint_{\mathcal{Y} \times \mathcal{Y}} k^*(\boldsymbol{y}, \boldsymbol{y}') x(\boldsymbol{y}) x(\boldsymbol{y}') d\boldsymbol{y} d\boldsymbol{y}'\right] \mathscr{D}x(\boldsymbol{y}), \quad (5)$$

where $\mathcal{K}$ and $\int_{\mathcal{Y}} k^*(\boldsymbol{y}, \boldsymbol{y}') \cdot d\boldsymbol{y}'$ are the integral operator with kernel function $k(\boldsymbol{y}, \boldsymbol{y}')$ and its inverse operator, respectively: $\mathcal{K}^{(*)} x(\boldsymbol{y}) = \int_{\mathcal{Y}} k^{(*)}(\boldsymbol{y}, \boldsymbol{y}') x(\boldsymbol{y}') d\boldsymbol{y}'$, $\mathcal{K}k^*(\boldsymbol{y}, \boldsymbol{y}') = \delta(\boldsymbol{y} - \boldsymbol{y}')$. Here $|\mathcal{K}|$ represents the function determinant [15] of $\mathcal{K}$, defined by the product of its eigenvalues. Using the relation (3) and the representation (5), we write the posterior of APP (1-2) in the following functional form,

$$p(x(\boldsymbol{y})|\mathcal{D}) \mathscr{D}x = \frac{1}{p(\mathcal{D})} \exp\left[-S\big(x(\boldsymbol{y}), \underline{x}(\boldsymbol{y})\big)\right] \mathscr{D}x, \quad (6)$$

where $S\big(x(\boldsymbol{y}), \underline{x}(\boldsymbol{y})\big)$ is the *action integral*, defined by

$$S\big(x(\boldsymbol{y}), \underline{x}(\boldsymbol{y})\big) = \int_{\mathcal{Y}} \left[\frac{1}{2} x(\boldsymbol{y}) \underline{x}(\boldsymbol{y}) + \rho(\boldsymbol{y}) x^2(\boldsymbol{y}) - 2 \sum_{n=1}^{N} \log x(\boldsymbol{y}) \delta(\boldsymbol{y} - \boldsymbol{y}_n)\right] d\boldsymbol{y} + \frac{1}{2} \log |\mathcal{K}|, \quad (7)$$

$\boldsymbol{y}_n = \boldsymbol{y}(\boldsymbol{t}_n)$, and $\underline{x}(\boldsymbol{y}) = \int_{\mathcal{Y}} k^*(\boldsymbol{y}, \boldsymbol{y}') x(\boldsymbol{y}') d\boldsymbol{y}'$. Then we apply calculus of variations to the action integral, where the functional derivative of $S\big(x(\boldsymbol{y}), \underline{x}(\boldsymbol{y})\big)$ on the MAP estimator $\hat{x}(\boldsymbol{y})$ should be equal to zero: $\frac{\delta S}{\delta \hat{x}(\boldsymbol{y})} \delta \hat{x}(\boldsymbol{y}) + \frac{\delta S}{\delta \underline{\hat{x}}(\boldsymbol{y})} \delta \underline{\hat{x}}(\boldsymbol{y}) = 0$, which results in the exact MAP estimator in a feasible form,

$$\hat{x}(\boldsymbol{y}) = 2 \sum_{n=1}^{N} h(\boldsymbol{y}, \boldsymbol{y}_n) v_n, \quad v_n = \hat{x}(\boldsymbol{y}_n)^{-1}, \quad (8)$$

where $h(\boldsymbol{y}, \boldsymbol{y}')$ is a transformed kernel function defined by a Fredholm integral equation of the second kind [32],

$$h(\boldsymbol{y}, \boldsymbol{y}') + 2 \int_{\mathcal{Y}} k(\boldsymbol{y}, \boldsymbol{s}) \rho(\boldsymbol{s}) h(\boldsymbol{s}, \boldsymbol{y}') d\boldsymbol{s} = k(\boldsymbol{y}, \boldsymbol{y}'). \quad (9)$$

See the supplementary material (§1) for the detailed derivations. Equation (8) shows that the MAP estimator of augmented permanental process involves the representer theorem under the transformed kernel function $h(\boldsymbol{y}, \boldsymbol{y}')$, and thus the Bayesian estimation reduces to a finite-dimensional optimization problem. Here, we call $h(\boldsymbol{y}, \boldsymbol{y}')$ the *equivalent kernel* following studies by Flaxman et al. [13] and Walder & Bishop [47]. Given the equivalent kernel, the unknown coefficients $v_n$ in (8) solve the simultaneous quadratic equations derived from (8),

$$\Delta_n \triangleq 2\, v_n \sum_{n'=1}^{N} h(\boldsymbol{y}_n, \boldsymbol{y}_{n'})v_{n'} - 1 = 0, \quad n = 1, 2, \ldots, N. \tag{10}$$

In this paper, we estimate a set of coefficients, $\{v_n\}_{n=1}^{N}$, by solving a minimization problem of the mean of the squared residuals, $\sum_{n=1}^{N} |\Delta_n|^2/N$, with a popular gradient descent algorithm, *Adam* [25]. We discuss how to obtain $h(\boldsymbol{y}, \boldsymbol{y}')$ in Section 2.3.

*Computational Complexity*

The objective function to be minimized in MAP estimation is $\sum_{n=1}^{N} |\Delta_n|^2/N$, which, in gradient descent algorithms, naively demands the computation of $\mathcal{O}(N^2)$ for each evaluation. However, when the equivalent kernel is given in degenerate form with rank $M$ ($< N$) such that $h(\boldsymbol{y}, \boldsymbol{y}') = \sum_{m=1}^{M} \varphi_m(\boldsymbol{y})\varphi_m(\boldsymbol{y}')$, the computational complexity reduces to $\mathcal{O}(NM)$, a linear computation against data size $N$. See Section 2.3 for the construction of the equivalent kernel $h(\boldsymbol{y}, \boldsymbol{y}')$.

## 2.3 Construction of Equivalent Kernels

### 2.3.1 Naive Approach

The equivalent kernel $h(\boldsymbol{y}, \boldsymbol{y}')$ solves the Fredholm integral equation (9), which includes the weight distribution of covariate, $\rho(\boldsymbol{y})$. Although it is possible to evaluate $\rho(\boldsymbol{y})$ numerically, we take an approach that avoids to treat $\rho(\boldsymbol{y})$ directly. Through the relation (4), we rewrite the integral term in (9) as $\int_{\mathcal{Y}} k(\boldsymbol{y}, \boldsymbol{s})\rho(\boldsymbol{y})h(\boldsymbol{s}, \boldsymbol{y}')d\boldsymbol{s} = \int_{\mathcal{T}} k(\boldsymbol{y}, \boldsymbol{y}(\boldsymbol{t}))h(\boldsymbol{y}(\boldsymbol{t}), \boldsymbol{y}')d\boldsymbol{t}$, and apply to it the *Nyström method* [30], a popular method to solve integral equations that approximates the integral operator by $J$-point numerical integration:

$$h(\boldsymbol{y}, \boldsymbol{y}') + 2\sum_{j=1}^{J} w_j k(\boldsymbol{y}, \boldsymbol{y}_j)h(\boldsymbol{y}_j, \boldsymbol{y}') = k(\boldsymbol{y}, \boldsymbol{y}'), \quad \boldsymbol{y}_j = \boldsymbol{y}(\boldsymbol{t}_j), \tag{11}$$

where $\boldsymbol{t}_j \in \mathcal{T}$ and $w_j \in \mathbb{R}$ represent the integration points and weights, respectively. It is easily shown that the equation (11) can be solved in an analytical form as follows,

$$h(\boldsymbol{y}, \boldsymbol{y}') = k(\boldsymbol{y}, \boldsymbol{y}') - \boldsymbol{k}(\boldsymbol{y})^\top \left(\boldsymbol{W}^{-1} + \boldsymbol{K}\right)^{-1}\boldsymbol{k}(\boldsymbol{y}'), \tag{12}$$

where $\boldsymbol{k}(\boldsymbol{y}) = (k(\boldsymbol{y}, \boldsymbol{y}_1), \ldots, k(\boldsymbol{y}, \boldsymbol{y}_J))^\top$, $\boldsymbol{W}_{jj'} = 2w_j\delta_{jj'}$, and $\boldsymbol{K}_{jj'} = k(\boldsymbol{y}_j, \boldsymbol{y}_{j'})$ for $1 \leq j, j' \leq J$. We call (12) the *naive approach* to constructing an equivalent kernel, and the complexity of evaluating $h(\boldsymbol{y}, \boldsymbol{y}')$ is $\mathcal{O}(J^3)$. In this paper, we use the quasi-Monte Carlo method [7], where $\{\boldsymbol{t}_j\}_{j=1}^{J}$ is a low-discrepancy sequence (Sobol sequence was used here) and $w_j = |\mathcal{T}|/J$ for the volume of the continuous domain $\mathcal{T}$. However, other techniques like Monte Carlo methods and a random quadrature approximation [50] are also possible.

### 2.3.2 Degenerate Approach

When the kernel function of GP has degenerate form with rank $M(< \infty)$,

$$k(\boldsymbol{y}, \boldsymbol{y}') = \sum_{m=1}^{M} \phi_m(\boldsymbol{y})\phi_m(\boldsymbol{y}') = \boldsymbol{\phi}(\boldsymbol{y})^\top\boldsymbol{\phi}(\boldsymbol{y}'), \tag{13}$$

it is well known that the Fredholm integral equation (9) can be solved analytically [3] as,

$$h(\boldsymbol{y}, \boldsymbol{y}') = \boldsymbol{\phi}(\boldsymbol{y})^\top(\boldsymbol{I}_M + 2\boldsymbol{A})^{-1}\boldsymbol{\phi}(\boldsymbol{y}'), \tag{14}$$

where $\phi(\boldsymbol{y}) = (\phi_1(\boldsymbol{y}), \phi_2(\boldsymbol{y}), \dots, \phi_M(\boldsymbol{y}))^\top$, and $\boldsymbol{A}$ is the $M \times M$ matrix defined by $\boldsymbol{A} = \int_{\mathcal{Y}} \rho(\boldsymbol{y})\phi(\boldsymbol{y})\phi(\boldsymbol{y})^\top d\boldsymbol{y}$. As in the naive approach, we rewrite and approximate the integral operator in $\boldsymbol{A}$ by using relation (4) and a $J$-point numerical integration, resulting in

$$\boldsymbol{A} = \sum_{j=1}^{J} w_j \phi(\boldsymbol{y}_j)\phi(\boldsymbol{y}_j)^\top, \quad \boldsymbol{y}_j = \boldsymbol{y}(\boldsymbol{t}_j), \tag{15}$$

where $\boldsymbol{t}_j \in \mathcal{T}$ and $w_j \in \mathbb{R}$ represent the integration points and weights, respectively. We call (14-15) the *degenerate approach* for constructing an equivalent kernel. The computational complexity of evaluating $h(\boldsymbol{y}, \boldsymbol{y}')$ is $\mathcal{O}(M^3 + JM^2)$, where $\mathcal{O}(JM^2)$ is given by numerical integration (15).

It should be emphasized here that when $k(\boldsymbol{y}, \boldsymbol{y}')$ has degenerate form with rank $M$, the relation (14) shows that the equivalent kernel $h(\boldsymbol{y}, \boldsymbol{y}')$ also has degenerate form obtained through Cholesky decomposition:

$$h(\boldsymbol{y}, \boldsymbol{y}') = (\boldsymbol{L}\phi(\boldsymbol{y}))^\top (\boldsymbol{L}\phi(\boldsymbol{y}')), \quad \boldsymbol{L}^\top \boldsymbol{L} = (\boldsymbol{I}_M + 2\boldsymbol{A})^{-1}. \tag{16}$$

The degenerate equivalent kernel (16) offers fast Bayesian estimation that scales linearly with $N$, the number of data points (see Section 2.2 and Section 2.4).

There are several ways of preparing degenerate kernel functions (13). *Random feature map* [33, 44, 45] approximates popular kernel functions, such as shift invariant kernels and additive kernels, by the inner products of randomized feature map vectors. Also, *Nyström approximation* for kernel methods [48, 49] finds a degenerate form of the function that best approximates an arbitrary kernel function at a finite subset of data points. We apply the two approximation methods to a Gaussian kernel in our experiments.

### 2.3.3 Discussion about Approximation Error of Integral Operator

We approximate the integral equation (9) through $J$-point numerical integration, which generally requires a large $J$ when the dimension of the covariate space, $\mathcal{Y}$, is high. However, it is worth noting that in our approaches, the required number of integration points, $J$, does not grow with the dimension of $\mathcal{Y}$ (covariate domain), but does with the dimension of $\mathcal{T}$ (observation domain). It is easily verified by the fact that the integral operator over $\mathcal{Y}$ in (9) can be transformed into the operator over $\mathcal{T}$ through the relation (4), and that the integration points are selected not on $\mathcal{Y}$ but on $\mathcal{T}$. Point processes are usually utilized to analyze at most three-dimensional data ($\mathcal{T} \subset \mathbb{R}^3$), and thus the required $J$ practically stays within a feasible size regardless of the dimension of $\mathcal{Y}$. An intuitive explanation for the requirement of $J$ is that observed covariate $\boldsymbol{y}(\boldsymbol{t})$ exists on a lower-dimensional manifold in high-dimensional domain $\mathcal{Y}$, and the effective volume of integral region is rather small.

The discussion so far is about the approximation error of the integral operator. Next, we discuss about the approximation error of the equivalent kernel under the approximate integral. Let the integral operator in (9) and its approximation through $J$-point numerical integration be denoted by $\mathcal{K}'$ and $\mathcal{K}'_J$, respectively. Then the equivalent kernel to solve (9) has the relation (for detailed derivation, see [3]),

$$h - h_J \propto (\mathcal{I} + 2\mathcal{K}'_J)^{-1}(\mathcal{K}'h - \mathcal{K}'_J h), \tag{17}$$

where $h$ and $h_J$ are the solutions of (9) under $\mathcal{K}'$ and $\mathcal{K}'_J$, respectively. The relation shows that the speed of convergence of $h_J$ to $h$ is at least as rapid as the speed of convergence of the integral operator, $\mathcal{K}'_J h$ to $\mathcal{K}'h$. Therefore, the required $J$ to obtain a good approximation of the equivalent kernel also stays within a feasible size, as long as the dimension of $\mathcal{T}$ is low.

### 2.4 Predictive Distribution and Marginal Likelihood

One of the advantages of GP models over non-Bayesian approaches is that they can provide predictive distributions and marginal likelihoods, which enable us to perform uncertainty evaluations, hyper-parameter optimization, and model selection in a Bayesian manner. We apply to APP (1-2) a Laplace approximation in the functional space, and find the approximate form of the predictive distribution and the marginal likelihood.

We now know the mode of the posterior, $\hat{x}(\boldsymbol{y})$, and consider a Taylor expansion of functional action potential $S(x(\boldsymbol{y}), \underline{x}(\boldsymbol{y}))$ centered on the mode such that

$$S(x(\boldsymbol{y}), \underline{x}(\boldsymbol{y})) \simeq S(\hat{x}(\boldsymbol{y}), \underline{\hat{x}}(\boldsymbol{y})) + \frac{1}{2}\iint_{\mathcal{Y} \times \mathcal{Y}} \sigma^*(\boldsymbol{y}, \boldsymbol{y}')(x(\boldsymbol{y}) - \hat{x}(\boldsymbol{y}))(x(\boldsymbol{y}') - \hat{x}(\boldsymbol{y}'))d\boldsymbol{y}d\boldsymbol{y}', \tag{18}$$

where $\sigma^*(\boldsymbol{y}, \boldsymbol{y}') = \frac{\delta^2 S(x,x)}{\delta x(\boldsymbol{y}) \delta x(\boldsymbol{y}')}\big|_{x=\hat{x}}$ is the second derivative of $S$. The first term in the Taylor expansion vanishes due to the stationary condition. The quadratic approximation of the action integral corresponds to the approximation of the posterior process by a GP, and the predictive covariance or the kernel function for the posterior GP, denoted by $\sigma(\boldsymbol{y}, \boldsymbol{y}')$, can be obtained by the functional inversion of $\sigma^*(\boldsymbol{y}, \boldsymbol{y}')$, which results in

$$\sigma(\boldsymbol{y}, \boldsymbol{y}') = h(\boldsymbol{y}, \boldsymbol{y}') - \boldsymbol{h}(\boldsymbol{y})^\top (\boldsymbol{Z} + \boldsymbol{H})^{-1} \boldsymbol{h}(\boldsymbol{y}'), \tag{19}$$

where

$$\boldsymbol{Z}_{nn'} = (2v_n^2)^{-1} \delta_{nn'}, \quad \boldsymbol{H}_{nn'} = h(\boldsymbol{y}_n, \boldsymbol{y}_{n'}), \quad \boldsymbol{h}(\boldsymbol{y}) = (h(\boldsymbol{y}, \boldsymbol{y}_1), \dots, h(\boldsymbol{y}, \boldsymbol{y}_N))^\top. \tag{20}$$

The full derivation of (19) is given in the supplementary material (§2). When the latent function $x(\boldsymbol{y})$ follows a posterior GP with mean of $\hat{x}(\boldsymbol{y})$ and the kernel $\sigma(\boldsymbol{y}, \boldsymbol{y}')$, it is easily verified that the value of the squared function, $\lambda = x^2(\boldsymbol{y})$, on each point of the covariate domain $\boldsymbol{y} \in \mathcal{Y}$ follows a Gamma distribution,

$$p_{\mu,\nu}(\lambda) = \frac{1}{\Gamma(\nu)\mu^\nu} \lambda^{\nu-1} \exp(-\lambda/\mu), \quad \mu = 2\sigma \frac{2\hat{x}^2 + \sigma}{\hat{x}^2 + \sigma}, \quad \nu = \frac{(\hat{x}^2 + \sigma)^2}{2\sigma(2\hat{x}^2 + \sigma)}, \tag{21}$$

where $\hat{x}$ and $\sigma$ are abbreviations of $\hat{x}(\boldsymbol{y})$ and $\sigma(\boldsymbol{y}, \boldsymbol{y})$, respectively. (21) represents the predictive distribution of the intensity as function of covariates.

Furthermore, under Laplace approximation (18), we can obtain the marginal likelihood, $p(\mathcal{D})$, in (6) by performing the path integral as,

$$\log p(\mathcal{D}) = \log \int \exp\left[-S\big(x(\boldsymbol{y}), \underline{x}(\boldsymbol{y})\big)\right] \mathscr{D}x \simeq -S\big(\hat{x}(\boldsymbol{y}), \hat{\underline{x}}(\boldsymbol{y})\big) + \frac{1}{2}\log|\Sigma|, \tag{22}$$

where $|\Sigma|$ is the functional determinant of integral operator $\Sigma = \int_{\mathcal{Y}} \cdot \sigma(\boldsymbol{y}, \boldsymbol{y}') d\boldsymbol{y}'$. We can rewrite the result in a more tractable form by substituting (7, 19) into (22):

$$\log p(\mathcal{D}) = \log|\boldsymbol{Z}| - \frac{1}{2}\log|\boldsymbol{I}_N + \boldsymbol{Z}^{-1}\boldsymbol{H}| + \frac{1}{2}\big(\log|\mathcal{H}| - \log|\mathcal{K}|\big) + (\log 2 - 1)N, \tag{23}$$

where $\boldsymbol{I}_N$ is the identity matrix with size $N$, and $|\mathcal{H}|$ is the functional determinant of integral operator $\mathcal{H} = \int_{\mathcal{Y}} \cdot h(\boldsymbol{y}, \boldsymbol{y}') d\boldsymbol{y}'$. The full derivation of (23) is given in the supplementary material (§3), and the method of estimating the functional determinant term is discussed in the end of this section.

*Computational Complexity*

Given the equivalent kernel $h(\boldsymbol{y}, \boldsymbol{y}')$, the evaluation of the predictive covariance (19) for a covariate pair, $(\boldsymbol{y}, \boldsymbol{y}')$, naively needs the computation of $\mathcal{O}(N^3 + MN^2)$ due to the matrix inversion in the second term. Fortunately, the complexity reduces to $\mathcal{O}(M^3 + NM^2)$ given a kernel with degenerate form, $h(\boldsymbol{y}, \boldsymbol{y}') = \sum_{m=1}^M \varphi_m(\boldsymbol{y})\varphi_m(\boldsymbol{y}')$, through the Woodbury matrix identity: The $N \times N$ matrix $\boldsymbol{H}$ can be decomposed into a product of $N \times M$ matrix $\boldsymbol{R}$, defined by $\boldsymbol{R}_{nm} = \varphi_m(\boldsymbol{y}_n)$, and its transpose as $\boldsymbol{H} = \boldsymbol{R}\boldsymbol{R}^\top$; the matrix inversion is transformed as $(\boldsymbol{Z} + \boldsymbol{R}\boldsymbol{R}^\top)^{-1} = \boldsymbol{Z}^{-1} - \boldsymbol{Z}^{-1}\boldsymbol{R}(\boldsymbol{I}_M + \boldsymbol{R}^\top\boldsymbol{Z}^{-1}\boldsymbol{R})^{-1}\boldsymbol{R}^\top\boldsymbol{Z}^{-1}$. Note that $\boldsymbol{Z}^{-1}\boldsymbol{R}$ costs $\mathcal{O}(NM)$ because $\boldsymbol{Z}$ is a diagonal matrix. Therefore, the computational complexity of the predictive distribution (21) is $\mathcal{O}(N^3 + MN^2)$ naively and $\mathcal{O}(M^3 + NM^2)$ with degenerate equivalent kernel.

In computing the marginal likelihood (23), the complexity of the first and the second terms is $\mathcal{O}(N^3 + MN^2)$ with a naive computation, but reduces to $\mathcal{O}(M^3 + NM^2)$ with degenerate equivalent kernel. The complexity of the third term, which is related to the functional determinants, depends on how the equivalent kernel $h(\boldsymbol{y}, \boldsymbol{y}')$ is constructed. Under the naive approach (12), the functional determinant of operator, $\mathcal{H} = \int_{\mathcal{Y}} \cdot h(\boldsymbol{y}, \boldsymbol{y}') d\boldsymbol{y}'$, can be obtained as a product of the functional determinant of $\mathcal{K}$ and a $J \times J$ matrix determinant: $|\mathcal{H}| = |\mathcal{K}||\boldsymbol{I}_J + \boldsymbol{W}\boldsymbol{K}|^{-1}$ (see the supplementary material (§4)). Thus the functional determinant term in (23) is given by the tractable determinant: $(\log|\mathcal{H}| - \log|\mathcal{K}|)/2 = -1/2\log|\boldsymbol{I}_J + \boldsymbol{W}\boldsymbol{K}|$, of which computational complexity is $\mathcal{O}(J^3 + MJ^2)$. Under the degenerate approach (14), the functional determinant of operator $\mathcal{H}$ can be obtained as a product of the functional determinant of $\mathcal{K}$ and a $M \times M$ matrix determinant: $|\mathcal{H}| = |\mathcal{K}||\boldsymbol{I}_M + 2\boldsymbol{A}|^{-1}$ (see the supplementary material (§4)). Thus the functional determinant term in (23) is given by the tractable determinant: $(\log|\mathcal{H}| - \log|\mathcal{K}|)/2 = -1/2\log|\boldsymbol{I}_M + 2\boldsymbol{A}|$, of which computational complexity is $\mathcal{O}(M^3 + JM^2)$, where $\mathcal{O}(JM^2)$ is given by numerical integration (15).

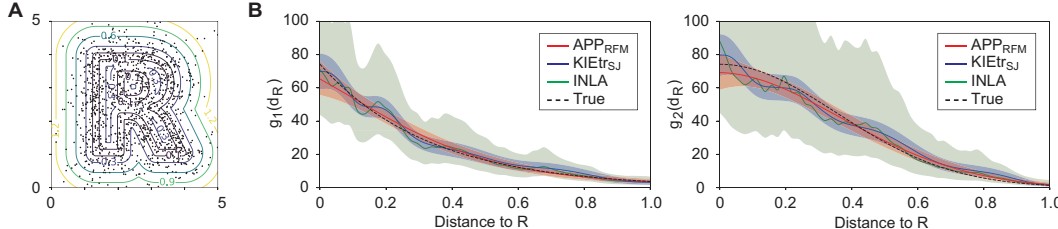

Figure 1: (A) Simulated data on $g_1(d_R)$ with $\alpha = 0.5$ and contours of distance to letter "R". (B) The estimated intensity functions. Solid lines and shaded areas represent predictive medians and [0.025, 0.975] prediction intervals, respectively.

## 3   Related Work

**Kernel Intensity Estimators** (KIEs) are the *de facto* standard nonparametric approaches for estimating point process intensities as functions of covariates in a non-Bayesian manner [17, 18, 5]. They construct intensity estimators by rescaling well-studied kernel (probability) density estimators for biased sample data. KIEs offer the lowest computational complexity as well as sophisticated methods for bandwidth selection. Analogue estimators, denoted by LLEs, have been also proposed based on local likelihood density estimators [5]. KIEs and LLEs have a variety of variants, some of which are readily available from library `spatstat` in R [4]. It is notable that the estimators also provide approximate point-wise confidence intervals for estimated intensities, which allows for uncertainty analysis.

**Gaussian Cox Processes** (GCPs) are the Bayesian alternatives to KIEs, where latent intensities are modeled by Gaussian processes or Gaussian random fields through positive link functions [29]. They provide us a methodology to perform the interval estimation of intensities as well as the hyper-parameter optimization in a Bayesian manner. GCPs have been intensively developed for the problem of estimating intensities as functions of each point on the observation space [1, 2, 9, 12, 11, 19, 23, 26, 36, 42], being a simpler case ($y(t) = t$) of what we consider in this paper. The estimation of intensities as functions of general covariates, which varies across observation domain, has recently been tackled by using domain discretization [21, 22, 28, 37, 51]; it partitions the observation domain into finite disjoint sub-areas and assumes the intensity to be constant over each sub-area. Although domain discretization is applicable to general GCPs regardless of link/kernel functions, it suffers from poor scaling in terms of domain dimension, as well as being sensitive to discretization size. Fitting GCPs to data are often based on time-consuming Markov chain Monte Carlo methods, but the integrated nested Laplace approximation (INLA) [21, 22, 37] can alleviate the fitting costs substantially. Fortunately, INLA is also accessible through the library `INLA` in R. INLA and KIE seem to be old methods published in 2008-2012, but to the best of our knowledge, there has been no feasible methods proposed to address the problem "estimating intensity as function of covariates" other than the methods. Thus, they are SOTA baselines for the problem we tackled.

**Permanental Processes** (PPs) are special GCPs that adopt quadratic forms of link functions [27], of which computational advantages have been highlighted by recent researches [13, 23, 26, 47]. The fact that PPs involve a representer theorem when $y(t) = t$ was exploited through the RKHS theory [13], the Mercer's theorem [47], and the path integral formulation [24]. Therefore, our proposed model is a generalization of their works for $y(t) \neq t$.

**Neural Network-based Point Processes** to explorer the dependence of point patterns on covariates have rarely been proposed. An exception is the deep mixture point process [31], which locates the finite representative points on the observation domain and models the intensity value on the point by a neural network model that eats as input the covariates on the point. This model provides a feasible estimation algorithm, but cannot perform the interval estimation of intensities.

## 4   Experiments

We examined the validity and the potential efficiency of our proposed model by evaluating it against the state-of-the-art Bayesian and non-Bayesian approaches on synthetic and open real-world data. As non-Bayesian approaches, we adopted KIEs (KIE) and LLEs (LLE) with the ratio (ra), the

Table 1: Results on two types of synthetic data across 100 trials with standard errors in brackets. $cpu$ is the CPU time in second, and $\tilde{N}$ is the average data size. The underlines represent the best predictive performances on each metric, and the performances not significantly ($p < 10^{-2}$) different from the best one under the Mann-Whitney U test with Holm method [20] are shown in bold.

| $g_1(d_R)$ | $\alpha = 0.2$ ($\tilde{N} = 232$) | | | | $\alpha = 0.5$ ($\tilde{N} = 586$) | | | | $\alpha = 1.0$ ($\tilde{N} = 1180$) | | | |
|---|---|---|---|---|---|---|---|---|---|---|---|---|
| | $l_{.025}$ | $l_{.5}$ | $l_{.975}$ | $cpu$ | $l_{.025}$ | $l_{.5}$ | $l_{.975}$ | $cpu$ | $l_{.025}$ | $l_{.5}$ | $l_{.975}$ | $cpu$ |
| KIEtr$_{SJ}$ | 0.15 (0.05) | **1.16** (0.35) | 0.21 (0.10) | 0.05 (0.00) | 0.29 (0.20) | 2.26 (0.52) | 0.36 (0.10) | 0.06 (0.00) | 0.43 (0.26) | 3.45 (0.84) | 0.51 (0.12) | 0.06 (0.00) |
| LLEra | 0.17 (0.09) | 1.41 (0.39) | 0.23 (0.05) | 0.07 (0.00) | 0.43 (0.34) | 2.91 (0.80) | 0.57 (0.29) | 0.08 (0.00) | 1.11 (0.85) | 5.05 (1.11) | 0.92 (0.45) | 0.08 (0.00) |
| INLA | 0.27 (0.03) | 1.31 (0.30) | 0.62 (0.13) | 101 (6.01) | 0.54 (0.06) | 2.54 (0.51) | 1.00 (0.14) | 108 (5.19) | 0.89 (0.07) | 3.71 (0.70) | 1.51 (0.20) | 110 (6.10) |
| APP$_{NAI}$ | **0.14** (0.06) | **1.08** (0.32) | **0.18** (0.09) | 9.30 (0.52) | **0.24** (0.10) | **1.93** (0.63) | **0.34** (0.22) | 14.1 (0.80) | **0.38** (0.26) | **2.74** (0.78) | **0.45** (0.33) | 25.5 (1.05) |
| APP$_{NYS}$ | **0.14** (0.05) | **1.08** (0.31) | **0.18** (0.09) | 0.61 (0.09) | **0.24** (0.10) | **1.93** (0.62) | **0.34** (0.22) | 2.62 (0.19) | **0.38** (0.27) | **2.72** (0.78) | **0.44** (0.33) | 2.81 (0.35) |
| APP$_{RFM}$ | **0.13** (0.07) | **1.06** (0.33) | **0.18** (0.13) | 2.42 (0.17) | **0.24** (0.10) | **1.90** (0.60) | **0.34** (0.24) | 2.48 (0.07) | **0.37** (0.27) | **2.65** (0.75) | **0.43** (0.34) | 2.56 (0.05) |
| APP$'_{RFM}$ | 0.17 (0.04) | 1.38 (0.32) | 0.24 (0.07) | 0.30 (0.01) | 0.31 (0.12) | 2.48 (0.64) | 0.45 (0.23) | 0.32 (0.01) | 0.49 (0.24) | 3.66 (0.82) | 0.60 (0.29) | 0.35 (0.01) |

| $g_2(d_R)$ | $\alpha = 0.2$ ($\tilde{N} = 322$) | | | | $\alpha = 0.5$ ($\tilde{N} = 801$) | | | | $\alpha = 1.0$ ($\tilde{N} = 1610$) | | | |
|---|---|---|---|---|---|---|---|---|---|---|---|---|
| | $l_{.025}$ | $l_{.5}$ | $l_{.975}$ | $cpu$ | $l_{.025}$ | $l_{.5}$ | $l_{.975}$ | $cpu$ | $l_{.025}$ | $l_{.5}$ | $l_{.975}$ | $cpu$ |
| KIEtr$_{SJ}$ | 0.18 (0.05) | 1.47 (0.35) | 0.24 (0.10) | 0.06 (0.00) | 0.38 (0.20) | 2.76 (0.52) | 0.40 (0.10) | 0.06 (0.00) | 0.83 (0.26) | 4.77 (0.84) | 0.64 (0.12) | 0.06 (0.00) |
| LLEra | 0.24 (0.09) | 1.67 (0.39) | 0.28 (0.05) | 0.07 (0.00) | 0.74 (0.34) | 3.47 (0.80) | 0.54 (0.29) | 0.08 (0.00) | 1.95 (0.85) | 6.17 (1.11) | 1.02 (0.45) | 0.08 (0.00) |
| INLA | 0.45 (0.03) | 1.73 (0.30) | 1.34 (0.13) | 112 (6.01) | 0.96 (0.06) | 3.31 (0.51) | 2.22 (0.14) | 115 (5.19) | 1.68 (0.07) | 5.04 (0.70) | 3.45 (0.20) | 116 (6.10) |
| APP$_{NAI}$ | **0.15** (0.06) | **1.06** (0.32) | **0.17** (0.09) | 10.1 (0.52) | **0.25** (0.10) | **1.87** (0.63) | **0.27** (0.22) | 17.7 (0.80) | **0.40** (0.26) | **2.75** (0.78) | **0.44** (0.33) | 40.1 (1.05) |
| APP$_{NYS}$ | **0.15** (0.05) | **1.08** (0.31) | **0.18** (0.09) | 1.02 (0.09) | **0.25** (0.10) | **1.88** (0.62) | **0.27** (0.22) | 2.72 (0.19) | **0.39** (0.27) | **2.79** (0.78) | **0.44** (0.33) | 3.15 (0.35) |
| APP$_{RFM}$ | **0.14** (0.07) | **0.99** (0.33) | **0.17** (0.13) | 2.40 (0.17) | **0.22** (0.10) | **1.76** (0.60) | **0.26** (0.24) | 2.57 (0.07) | **0.36** (0.27) | **2.53** (0.75) | **0.40** (0.34) | 2.78 (0.05) |
| APP$'_{RFM}$ | 0.20 (0.04) | 1.57 (0.32) | 0.28 (0.07) | 0.30 (0.01) | 0.37 (0.12) | 2.76 (0.64) | 0.46 (0.23) | 0.34 (0.01) | 0.57 (0.24) | 4.06 (0.82) | 0.65 (0.29) | 0.40 (0.01) |

reweighted (re), and the transformation (tr) forms [5], and selected the kernel bandwidths of KIEs by Silverman's rule of thumb [40] (SI) and a method by Sheather & Jones [39] (SJ), resulting in the nine non-Bayesian variants (e.g. KIEtr$_{SJ}$). Some of them are not implemented for multivariate covariates in `spatstat`, and they were applied only to synthetic data, where $\mathcal{Y} \subset \mathbb{R}$. As a Bayesian approach, we employed the log-Gaussian Cox process with INLA [22] (INLA) with $100 \times 100$ domain discretization, where the random walk model of order 1 was used as the prior process, and the hyperparameters were optimized by maximizing the marginal likelihood through the 16-points grid search. For our proposal, we adopted APPs with the naive approach (APP$_{NAI}$) and with the degenerate approaches of the random Fourier map [33] (APP$_{RFM}$) and the Nyström approximation [49] (APP$_{NYS}$) [2]. We applied to the APPs a multiplicative Gaussian kernel, $k(\boldsymbol{y}, \boldsymbol{y}') = \theta_0 \prod_{d=1}^{D_y} e^{-(\theta_d(y_d - y'_d))^2}$, where the hyper-parameter $\theta = (\theta_0, \ldots, \theta_{D_y})$ was optimized for each data by maximizing the marginal likelihood (23) through the 25-points grid search. A MacBook Pro with 4-core CPU (2.8 GHz Intel Core i7) was used. For details of the model configurations, see the supplementary material (§5).

## 4.1 Synthetic Data

Following [5], we created 2D data sets ($\mathcal{T} \subset \mathbb{R}^2$, $\mathcal{Y} \subset \mathbb{R}$) generated from two types of intensity functions: $\lambda(\boldsymbol{t}) = g_1(d_R(\boldsymbol{t})) = \alpha \exp(5 - 3d_R(\boldsymbol{t}))$ and $\lambda(\boldsymbol{t}) = g_2(d_R(\boldsymbol{t})) = \alpha \exp(5 - 4d_R(\boldsymbol{t})^2)$, each of which has 100 trial sequences. Here $d_R(\boldsymbol{t})$ denotes the shortest distance from a given location $\boldsymbol{t}$ to the set of lines arranged in the shape of the letter "R" (see Figure 1A). The coefficient

---

[2] Code and data are provided at `https://github.com/HidKim/APP`.

Table 2: Results on real-world data across 10 trials. $cpu$ is the CPU time in second, and the underlines represent the best predictive performances on each metric. Notations follow Table 1.

| | copper | | | bei | | | clmfires | | |
|---|---|---|---|---|---|---|---|---|---|
| | $ll_{\text{test}}$ | $cl_{\text{test}}$ | $cpu$ | $ll_{\text{test}}$ | $cl_{\text{test}}$ | $cpu$ | $ll_{\text{test}}$ | $cl_{\text{test}}$ | $cpu$ |
| KIEra$_{\text{SI}}$ | **5.65** (0.37) | **1.64** (0.09) | 0.32 (0.01) | 2.92 (0.00) | 1.31 (0.07) | 0.26 (0.00) | 2.39 (0.00) | 1.30 (0.08) | 0.24 (0.01) |
| KIEre$_{\text{SI}}$ | **5.65** (0.37) | **1.64** (0.09) | 0.33 (0.01) | 2.92 (0.00) | 1.31 (0.07) | 0.25 (0.00) | 2.40 (0.00) | 1.38 (0.08) | 0.24 (0.01) |
| INLA | 5.64 (0.34) | 1.66 (0.10) | 82.9 (3.00) | 2.90 (0.01) | 1.13 (0.06) | 131 (3.37) | 2.39 (0.01) | **1.18** (0.06) | 129 (4.61) |
| APP$_{\text{NYS}}$ | **5.65** (0.37) | 1.64 (0.09) | 0.60 (0.06) | 2.87 (0.01) | 1.01 (0.05) | 15.8 (0.41) | 2.36 (0.01) | 1.15 (0.07) | 17.9 (1.06) |
| APP$_{\text{RFM}}$ | **5.65** (0.37) | **1.64** (0.09) | 2.12 (0.10) | **2.88** (0.01) | **1.02** (0.05) | 4.64 (0.21) | **2.36** (0.01) | **1.16** (0.07) | 4.28 (0.22) |

$\alpha$ was set as 0.2, 0.5, and 1.0. The predictive performance was evaluated based on the integrated $\rho$-quantile loss [38], defined as $l_\rho \triangleq \int_{\mathcal{Y}} 2\big(g(y)-\hat{g}(y)\big)\big(\rho\mathrm{I}_{g(y)>\hat{g}(y)} - \big(1-\rho\big)\mathrm{I}_{g(y)\leq\hat{g}(y)}\big)dy$, where I, $\hat{g}(y)$, and $g(y)$ denote the indicator, the predicted $\rho$-quantile of the intensity function on covariate domain, and the true one, respectively. Here, we adopted $l_{.025}$, $l_{.5}$ (integrated absolute error) and $l_{.975}$.

Table 1 displays the predictive performances on various synthetic data, where only the methods with relatively good accuracy are listed due to space limitations (see the supplementary material (§5) for the full results). It shows that our approaches (APPs) outperformed the reference methods across all predictive metrics, where the performance gaps were not marginal (e.g., the improvements of $l_{.5}$ were 9%~45%). Furthermore, APPs with the degenerate approaches (APP$_{\text{NYS}}$ and APP$_{\text{RFM}}$) were performed several tens of times faster than INLA, a feasible Bayesian method based on domain discretization. Table 1 empirically suggests that our Laplace approximation on the functional space (see Section 2.4) can provide a reasonable approximation of the predictive distribution: APPs' good scores regarding $l_{.025}$ and $l_{.975}$ demonstrate that the approximation can recover the posterior distribution accurately; APP optimized with the approximate marginal likelihood (APP$_{\text{RFM}}$) outperformed APP with an initial hyper-parameter (APP$'_{\text{RFM}}$), showing the practical utility of the approximate marginal likelihood for determining hyper-parameters. Figure 1B implies that INLA tends to overestimate the predictive errors. Also, we ran an additional experiment based on larger data sets and synthetic data sets with 2D covariate (see the supplementary material (§6-§7)).

## 4.2 Real-world Data

We examined the validity of our approach based on real-world spatial ($\mathcal{T} \subset \mathbb{R}^2$) data provided by spatstat.data in R (GPL-3) [4]: copper consists of 67 location points of copper ore deposits and 146 line segments representing faults. The covariate of interest is the shortest distance from a given location $t$ to the set of faults ($\mathcal{Y} \subset \mathbb{R}$); bei consists of locations of 3605 trees of the species *Beilschmiedia pendula* and geo-information in a tropical rain forest. The covariates of interest are the terrain elevation and the terrain slope ($\mathcal{Y} \subset \mathbb{R}^2$); clmfires consists of locations of forest fires in the Castilla-La Mancha region of Spain and the geographical information. We restricted the analysis to events happening within a rectangular region (180, 60) – (330, 360), resulting in 4241 location points. The covariates of interest are the terrain elevation and the terrain slope ($\mathcal{Y} \subset \mathbb{R}^2$). For each data set, we randomly split the data points into 10 subsets, assigned one to test and the others to training data, and conducted 10-fold cross validations of the predictive performances based on the negative test log likelihood of point patterns ($ll_{\text{test}}$) and the negative test likelihood of counts ($cl_{\text{test}}$). Details of the metrics are given in the supplementary material (§5). Note that to eliminate the effect of data assignment bias, the evaluation multiplied estimated intensity by a factor, $N_{\text{test}}/N_{\text{train}}$.

Table 2 displays the predictive performances on the three real-world data sets, which shows that our approach (APP$_*$) outperformed the reference methods in most cases: On bei, APP achieved the significant 10%~20% improvements of $cl_{\text{test}}$ against the reference methods; On clmfires, the APP's improvements of predictive performance against INLA were small, but being statistically significant for $ll_{\text{test}}$; On copper, the predictive performances of all the methods were found not to be significantly different from each other, which might be due to the too small data size ($N = 67$).

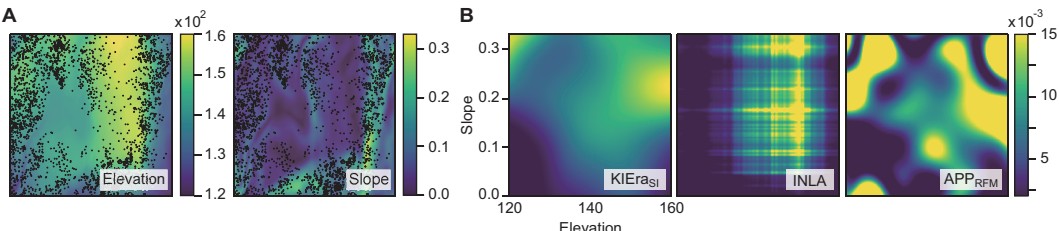

Figure 2: Estimation results on `bei`. (A) Covariate maps on 2D spatial domain with observed points (black dots). (B) The estimated intensities as functions of 2D covariates.

It should be noted that the CPU times of INLA are less affected by the size of data due to domain discretization. Thus INLA remains a valuable estimation approach when the size of data is large and the dimension of observation domain is low. Figure 2 displays the estimation results on `bei`, which implies that KIEs tend to underestimate the intensity modulation than Bayesian methods. Here, the estimated intensity by INLA seems less smooth than the others, which is due to the additive assumption in the R package `INLA`: $\lambda(\boldsymbol{t}) \propto \exp[g_1(\text{elev}(\boldsymbol{t})) + g_2(\text{slope}(\boldsymbol{t}))]$.

## 5    Conclusions

We have proposed a novel Bayesian estimation method of point process intensity as function of co-variates. We augmented a permanental process such that latent intensities can be defined on covariate space, and showed that a representer theorem holds for the augmented process. By exploiting the representer theorem, we have derived a fast estimation scheme which scales linearly with data size without using either domain discretization or MCMC. Based on synthetic and open real-world data, we confirmed that our method outperforms state-of-the-art methods in terms of predictive accuracy while being substantially faster than a conventional Bayesian method. Although the key derivation of the scheme are based on the methodology proposed in [24], we believe that the result makes a non-incremental impact on the Bayesian estimation community for point processes. Also, our finding makes a non-trivial contribution to the reproducing kernel Hilbert space community because it shows a novel and practically important likelihood function for which the representer theorem holds under the regularization term of the squared Hilbert space norm.

**Future work & limitations:** This time, we optimized the hyper-parameter of our model by maximizing the marginal likelihood through grid search, but it is clearly beneficial if gradient descent algorithms could be utilized. Actually, the maximization of the marginal likelihood, which has an argmin (see Section 2.2), belongs to a bi-level optimization, and we can compute gradients of the marginal likelihood regarding the hyper-parameter exactly [16]. Better hyper-parameters could be found by implementing the gradient descent methods.

We achieved a scalable algorithm under a degenerate form of equivalent kernel function, but an equivalent kernel function with a small rank might lead to overconfident predictive covariance [34]. A promising approach to the problem is employing variational Bayesian (VB) approximations with inducing points, which usually provide a full-rank predictive covariance. VB-based methods for the scenario "estimating intensity as function of covariates" have not been proposed so far, and thus it is an important next step of our study.

As in ordinary permanental processes, the nodal line problem could arise in the augmented perma-nental process (APP): the posterior distribution of the latent variable $x(\cdot)$ has many local modes since $\pm x(\cdot)$ can lead to similar intensity functions $\lambda = x^2(\cdot)$, and artificial zero crossings of $x(\cdot)$ could happen especially on a location where the intensity is row. To alleviate the problem, John and Hensman [23] have proposed to extend the quadratic link function to include an offset parameter $\beta$, so that $k(x) = (x + \beta)^2$, which is also valid for our APP.

One promising extension of our model is for the survival analysis methods, which has been actively developed in the medical field. In it, the observation domain is time (one-dimensional) and the observation is finished when an event occurs for the first time, to which our model cannot be applied directly due to the dependence of the observation domain on the event data. However, it might be possible to construct a feasible algorithm through the similar methodology proposed in our paper.

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
