# Supplementary Material
## "Fast Bayesian Estimation of Point Process Intensity as Function of Covariates"

**Hideaki Kim**     **Taichi Asami**     **Hiroyuki Toda**[*]
NTT Human Informatics Laboratories
NTT Corporation
{hideaki.kin.cn, taichi.asami.ka}@hco.ntt.co.jp, hirotoda@acm.org

## S1   Derivation of MAP Estimator

We detail here the derivation of the MAP estimator (8). The functional derivative of $S(x(\boldsymbol{y}), \underline{x}(\boldsymbol{y}))$ should be zero on the MAP estimator $\hat{x}(\boldsymbol{y})$:

$$\delta S\big(\hat{x}(\boldsymbol{y}), \underline{\hat{x}}(\boldsymbol{y})\big) = \int_{\mathcal{Y}} \left[ \frac{\delta S}{\delta \hat{x}(\boldsymbol{y})} \delta x(\boldsymbol{y}) + \frac{\delta S}{\delta \underline{\hat{x}}(\boldsymbol{y})} \delta \underline{x}(\boldsymbol{y}) \right] d\boldsymbol{y} + O((\delta x)^2)$$

$$\simeq \int_{\mathcal{Y}} \left[ 2\rho(\boldsymbol{y})\hat{x}(\boldsymbol{y}) - \sum_{n=1}^{N} \frac{2}{\hat{x}(\boldsymbol{y}_n)} \delta(\boldsymbol{y} - \boldsymbol{y}_n) + \frac{1}{2} \underline{\hat{x}}(\boldsymbol{y}) \right] \delta x d\boldsymbol{y} + \int_{\mathcal{Y}} \frac{1}{2} \hat{x}(\boldsymbol{y}) \delta \underline{x} d\boldsymbol{y}$$

$$= \int_{\mathcal{Y}} \left[ 2\rho(\boldsymbol{y})\hat{x}(\boldsymbol{y}) - \sum_{n=1}^{N} \frac{2}{\hat{x}(\boldsymbol{y}_n)} \delta(\boldsymbol{y} - \boldsymbol{y}_n) + \underline{\hat{x}}(\boldsymbol{y}) \right] \delta x d\boldsymbol{y} = 0,$$

where the following relation was used,

$$\int_{\mathcal{Y}} \hat{x}(\boldsymbol{y}) \delta \underline{x} d\boldsymbol{y} = \int_{\mathcal{Y}} \hat{x}(\boldsymbol{y}) \int_{\mathcal{Y}} k^*(\boldsymbol{y}, \boldsymbol{y}') \delta x(\boldsymbol{y}') d\boldsymbol{y}' d\boldsymbol{y}$$

$$= \int_{\mathcal{Y}} d\boldsymbol{y}' \delta x(\boldsymbol{y}') \int_{\mathcal{Y}} k^*(\boldsymbol{y}, \boldsymbol{y}') \hat{x}(\boldsymbol{y}) d\boldsymbol{y}$$

$$= \int_{\mathcal{Y}} \underline{\hat{x}}(\boldsymbol{y}') \delta x d\boldsymbol{y}'. \quad \because) \; k^*(\boldsymbol{y}, \boldsymbol{y}') = k^*(\boldsymbol{y}', \boldsymbol{y})$$

Thus the following equation is derived,

$$\underline{\hat{x}}(\boldsymbol{y}) + 2\rho(\boldsymbol{y})\hat{x}(\boldsymbol{y}) = \sum_{n=1}^{N} \frac{2}{\hat{x}(\boldsymbol{y}_n)} \delta(\boldsymbol{y} - \boldsymbol{y}_n), \quad \boldsymbol{y} \in \mathcal{Y}. \tag{S1}$$

By applying operator $\mathcal{K}$ to (S1), we obtain a linear integral equation that derives the MAP estimator $\hat{x}(\boldsymbol{y})$ as follows,

$$\hat{x}(\boldsymbol{y}) + 2 \int_{\mathcal{Y}} k(\boldsymbol{y}, \boldsymbol{y}') \rho(\boldsymbol{y}') \hat{x}(\boldsymbol{y}') d\boldsymbol{y}' = 2 \sum_{n=1}^{N} k(\boldsymbol{y}, \boldsymbol{y}_n) \hat{x}(\boldsymbol{y}_n)^{-1}, \quad \boldsymbol{y} \in \mathcal{Y}.$$

The linearity of the integral equation permits a representation of the form

$$\hat{x}(\boldsymbol{y}) = 2 \sum_{n=1}^{N} h(\boldsymbol{y}, \boldsymbol{y}_n) \hat{x}(\boldsymbol{y}_n)^{-1}, \tag{S2}$$

36th Conference on Neural Information Processing Systems (NeurIPS 2022).

---

[*]Current affiliation is Yokohama City University.

where $h(\boldsymbol{y}, \boldsymbol{y}')$ is a positive semi-definite kernel defined by the following integral equation,

$$h(\boldsymbol{y}, \boldsymbol{y}') + 2 \int_{\mathcal{Y}} k(\boldsymbol{y}, \boldsymbol{s}) \rho(\boldsymbol{s}) h(\boldsymbol{s}, \boldsymbol{y}') d\boldsymbol{s} = k(\boldsymbol{y}, \boldsymbol{y}'). \tag{S3}$$

## S2 Derivation of Predictive Covariance

We detail the derivation of the predictive covariance shown in (19-20). The predictive inverse co-variance (precision), denoted by $\sigma^*(\boldsymbol{y}, \boldsymbol{y}')$, is given by the second functional derivative of $S$, which is written as

$$\sigma^*(\boldsymbol{y}, \boldsymbol{y}') = \frac{\delta^2 S(x, \underline{x})}{\delta x(\boldsymbol{y}) \delta x(\boldsymbol{y}')} \bigg|_{x=\hat{x}} = z^*(\boldsymbol{y}, \boldsymbol{y}') + h^*(\boldsymbol{y}, \boldsymbol{y}'),$$

where

$$\begin{aligned}
z^*(\boldsymbol{y}, \boldsymbol{y}') &= 2 \sum_{n=1}^{N} \hat{x}(\boldsymbol{y}_n)^{-2} \delta(\boldsymbol{y} - \boldsymbol{y}_n) \delta(\boldsymbol{y}' - \boldsymbol{y}_n), \\
h^*(\boldsymbol{y}, \boldsymbol{y}') &= 2\rho(\boldsymbol{y}') \delta(\boldsymbol{y} - \boldsymbol{y}') + k^*(\boldsymbol{y}, \boldsymbol{y}').
\end{aligned} \tag{S4}$$

Let the integral operators corresponding to $\sigma(\boldsymbol{y}, \boldsymbol{y}')$, $z(\boldsymbol{y}, \boldsymbol{y}')$, and $h(\boldsymbol{y}, \boldsymbol{y}')$ be denoted by $\Sigma$, $\mathcal{Z}$, and $\mathcal{H}$, respectively, and their inverse counterparts by $^*$. Using the fact that operator $\mathcal{Z}^*$ is factorized as,

$$\mathcal{Z}^* = \int_{\mathcal{Y}} \cdot z^*(\boldsymbol{y}, \boldsymbol{y}') d\boldsymbol{y}' = \mathcal{U}^\top \boldsymbol{Z}^{-1} \mathcal{U},$$

$$\boldsymbol{Z}_{nn'} = 2^{-1} \hat{x}(\boldsymbol{y}_n)^2 \delta_{nn'}, \quad \mathcal{U}_n = \int_{\mathcal{Y}} \cdot \delta(\boldsymbol{y}' - \boldsymbol{y}_n) d\boldsymbol{y}',$$

we obtain the predictive covariance $\sigma(\boldsymbol{y}, \boldsymbol{y}')$ with a finite (thus tractable) $N$-dimensional matrix representation as follows:

$$\begin{aligned}
\Sigma &= \int_{\mathcal{Y}} \cdot \sigma(\boldsymbol{y}, \boldsymbol{y}') d\boldsymbol{y}' = (\mathcal{Z}^* + \mathcal{H}^*)^* = (\mathcal{U}^\top \boldsymbol{Z}^{-1} \mathcal{U} + \mathcal{H}^*)^* \\
&= \mathcal{H} - \mathcal{H} \mathcal{U}^\top (\boldsymbol{Z} + \mathcal{U} \mathcal{H} \mathcal{U}^\top)^* \mathcal{U} \mathcal{H} \\
&= \int_{\mathcal{Y}} \cdot \Big[ h(\boldsymbol{y}, \boldsymbol{y}') - \boldsymbol{h}(\boldsymbol{y})^\top (\boldsymbol{Z} + \boldsymbol{H})^{-1} \boldsymbol{h}(\boldsymbol{y}') \Big] d\boldsymbol{y}',
\end{aligned}$$

or equivalently,

$$\sigma(\boldsymbol{y}, \boldsymbol{y}') = h(\boldsymbol{y}, \boldsymbol{y}') - \boldsymbol{h}(\boldsymbol{y})^\top (\boldsymbol{Z} + \boldsymbol{H})^{-1} \boldsymbol{h}(\boldsymbol{y}'), \tag{S5}$$

where $\boldsymbol{H}_{nn'} = h(\boldsymbol{y}_n, \boldsymbol{y}_{n'})$, $\boldsymbol{h}(\boldsymbol{y}) = (h(\boldsymbol{y}, \boldsymbol{y}_1), \dots, h(\boldsymbol{y}, \boldsymbol{y}_N))^\top$; we used the Woodbury matrix identity in this derivation. Here, Equation (S4) states that the operators $\mathcal{H}$, $\mathcal{K}$, and $\mathcal{A} = 2 \int_{\mathcal{Y}} \cdot \rho(\boldsymbol{y}') \delta(\boldsymbol{y} - \boldsymbol{y}') d\boldsymbol{y}'$ hold the relation,

$$\mathcal{H}^* = \mathcal{A} + \mathcal{K}^* \iff (\mathcal{I} + \mathcal{K}\mathcal{A})\mathcal{H} = \mathcal{K},$$

which is equivalent to Equation (S3). Thus $h(\boldsymbol{y}, \boldsymbol{y}')$ in (S5) is equal to the equivalent kernel function defined by Equation (S3).

## S3 Derivation of Marginal Likelihood

We detail the derivation of the marginal likelihood, $p(\mathcal{D})$, shown in (23). Under Laplace approximation (18), we can obtain the marginal likelihood by performing path integral as follows:

$$\begin{aligned}
\log p(\mathcal{D}) &= \log \int \exp \big[ -S(x(\boldsymbol{y}), \underline{x}(\boldsymbol{y})) \big] \mathscr{D}x \\
&\simeq -S\big( \hat{x}(\boldsymbol{y}), \underline{\hat{x}}(\boldsymbol{y}) \big) + \log \int e^{-\frac{1}{2} \iint_{\mathcal{Y} \times \mathcal{Y}} \sigma^*(\boldsymbol{y}, \boldsymbol{y}')(x(\boldsymbol{y}) - \hat{x}(\boldsymbol{y}))(x(\boldsymbol{y}') - \hat{x}(\boldsymbol{y}')) d\boldsymbol{y} d\boldsymbol{y}'} \mathscr{D}x \\
&= -S\big( \hat{x}(\boldsymbol{y}), \underline{\hat{x}}(\boldsymbol{y}) \big) + \frac{1}{2} \log |\Sigma|,
\end{aligned}$$

where we used the relation in path integral (see Equation (9) in [3]),

$$\int \exp\Big[-\frac{1}{2}\iint_{\mathcal{Y}\times\mathcal{Y}}\sigma^*(\boldsymbol{y},\boldsymbol{y}')(x(\boldsymbol{y})-\hat{x}(\boldsymbol{y}))(x(\boldsymbol{y}')-\hat{x}(\boldsymbol{y}'))d\boldsymbol{y}d\boldsymbol{y}'\Big]\mathscr{D}x = \sqrt{|\varSigma|}.$$

Substituting (S1) into (7), we can write down $S(\hat{x}(\boldsymbol{y}),\hat{\underline{x}}(\boldsymbol{y}))$ as

$$S(\hat{x}(\boldsymbol{y}),\hat{\underline{x}}(\boldsymbol{y})) = \frac{1}{2}\log|\mathcal{K}| - 2\sum_{n=1}^{N}\log\hat{x}(\boldsymbol{y}_n) + \int_{\mathcal{Y}}\rho(\boldsymbol{y})\hat{x}(\boldsymbol{y})^2 d\boldsymbol{y} + \frac{1}{2}\int_{\mathcal{Y}}\hat{x}(\boldsymbol{y})\hat{\underline{x}}(\boldsymbol{y})d\boldsymbol{y}$$

$$= \frac{1}{2}\log|\mathcal{K}| - 2\sum_{n=1}^{N}\log\hat{x}(\boldsymbol{y}_n) + \int_{\mathcal{Y}}\rho(\boldsymbol{y})\hat{x}(\boldsymbol{y})^2 d\boldsymbol{y}$$

$$+ \frac{1}{2}\int_{\mathcal{Y}}\hat{x}(\boldsymbol{y})\Big[\sum_{n=1}^{N}\frac{2}{\hat{x}(\boldsymbol{y}_n)}\delta(\boldsymbol{y}-\boldsymbol{y}_n) - 2\rho(\boldsymbol{y})\hat{x}(\boldsymbol{y})\Big]d\boldsymbol{y}$$

$$= \frac{1}{2}\log|\mathcal{K}| - 2\sum_{n=1}^{N}\log\hat{x}(\boldsymbol{y}_n) + N$$

$$= \frac{1}{2}\log|\mathcal{K}| - \log|\boldsymbol{Z}| - (\log 2 - 1)N,$$

where $\boldsymbol{Z}_{nn'} = 2^{-1}\hat{x}(\boldsymbol{y}_n)^2\delta_{nn'}$. Furthermore, by using the matrix determinant lemma, we can rewrite $\log|\varSigma|$ as

$$\log|\varSigma| = \log|\mathcal{H} - \mathcal{H}\mathcal{U}^{\top}(\boldsymbol{Z}+\mathcal{U}\mathcal{H}\mathcal{U}^{\top})^*\mathcal{U}\mathcal{H}|$$

$$= \log|\mathcal{H}| - \log|\boldsymbol{Z}+\mathcal{U}\mathcal{H}\mathcal{U}^{\top}| + \log|(\boldsymbol{Z}+\mathcal{U}\mathcal{H}\mathcal{U}^{\top}) - (\mathcal{U}\mathcal{H})\mathcal{H}^*(\mathcal{H}\mathcal{U}^{\top})|$$

$$= \log|\mathcal{H}| - \log|\boldsymbol{Z}+\mathcal{U}\mathcal{H}\mathcal{U}^{\top}| + \log|\boldsymbol{Z}|$$

$$= \log|\mathcal{H}| - \log|\boldsymbol{I}_N + \boldsymbol{Z}^{-1}\boldsymbol{H}|.$$

Finally, we obtain the marginal likelihood in a tractable form,

$$\log p(\mathcal{D}) = \log|\boldsymbol{Z}| - \frac{1}{2}\log|\boldsymbol{I}_N + \boldsymbol{Z}^{-1}\boldsymbol{H}| + \frac{1}{2}\big(\log|\mathcal{H}| - \log|\mathcal{K}|\big) + (\log 2 - 1)N.$$

## S4   Functional Determinant of Equivalent Kernel

We detail the derivation of the functional determinant of equivalent kernel, $|\mathcal{H}|$, when the naive and degenerate approaches are applied.

### S4.1   Naive Approach

The equivalent kernel is constructed under the naive approach as follows:

$$h(\boldsymbol{y},\boldsymbol{y}') = k(\boldsymbol{y},\boldsymbol{y}') - \boldsymbol{k}(\boldsymbol{y})^{\top}(\boldsymbol{W}^{-1}+\boldsymbol{K})^{-1}\boldsymbol{k}(\boldsymbol{y}'),$$

where $\boldsymbol{k}(\boldsymbol{y}) = (k(\boldsymbol{y},\boldsymbol{y}_1),\ldots,k(\boldsymbol{y},\boldsymbol{y}_J))^{\top}$, $\boldsymbol{W}_{jj'} = 2w_j\delta_{jj'}$, and $\boldsymbol{K}_{jj'} = k(\boldsymbol{y}_j,\boldsymbol{y}_{j'})$ for $1 \le j,j' \le J$. Let the integral operators corresponding to $\boldsymbol{k}(\boldsymbol{y})$ and $\boldsymbol{k}(\boldsymbol{y})^{\top}$ be denoted by $\underline{\mathcal{K}} = \int_{\mathcal{Y}}\cdot\,\boldsymbol{k}(\boldsymbol{y}')d\boldsymbol{y}'$ and $\underline{\mathcal{K}}^{\top} = \int_{\mathcal{Y}}\cdot\,\boldsymbol{k}(\boldsymbol{y}')^{\top}d\boldsymbol{y}'$, respectively. Then we can rewrite $|\mathcal{H}|$ as

$$|\mathcal{H}| = |\mathcal{K} - \underline{\mathcal{K}}^{\top}(\boldsymbol{W}^{-1}+\boldsymbol{K})^{-1}\underline{\mathcal{K}}| = |\mathcal{K}||\boldsymbol{W}^{-1}+\boldsymbol{K}|^{-1}|\boldsymbol{W}^{-1}+\boldsymbol{K}-\underline{\mathcal{K}}\mathcal{K}^*\underline{\mathcal{K}}^{\top}|, \qquad \text{(S6)}$$

where $\mathcal{K}^*$ is the inverse operator of $\mathcal{K}$, and we used the matrix determinant lemma. Mercer's theorem [5] states that the kernel function and its inverse counterpart, $k(\boldsymbol{y},\boldsymbol{y}')$ and $k^*(\boldsymbol{y},\boldsymbol{y}')$, respectively, have diagonal representations,

$$k(\boldsymbol{y},\boldsymbol{y}') = \sum_{m=1}^{\infty}\nu_m e_m(\boldsymbol{y})e_m(\boldsymbol{y}'), \quad k^*(\boldsymbol{y},\boldsymbol{y}') = \sum_{m=1}^{\infty}\nu_m^{-1}e_m(\boldsymbol{y})e_m(\boldsymbol{y}'), \qquad \text{(S7)}$$

where $\{e_m(\cdot)\}_m$ is an orthonormal basis comprising eigenfunctions of $\mathcal{K}$, and $\{\nu_m\}_m$ is the set of eigenvalues of $\mathcal{K}$. Using the relation (S7), we obtain a more tractable form of $\underline{\mathcal{K}}\mathcal{K}^*\underline{\mathcal{K}}^\top$, the $J \times J$ matrix term appeared in (S6), as follows:

$$
\begin{aligned}
\left(\underline{\mathcal{K}}\mathcal{K}^*\underline{\mathcal{K}}^\top\right)_{jj'} &= \int_{\mathcal{Y}}\int_{\mathcal{Y}} \boldsymbol{k}_j(\boldsymbol{y})\boldsymbol{k}_{j'}(\boldsymbol{y}')k^*(\boldsymbol{y},\boldsymbol{y}')d\boldsymbol{y}d\boldsymbol{y}' \\
&= \int_{\mathcal{Y}}\int_{\mathcal{Y}}\left(\sum_{m=1}^{\infty}\nu_m e_m(\boldsymbol{y})e_m(\boldsymbol{y}_j)\right)\left(\sum_{m'=1}^{\infty}\nu_{m'}e_{m'}(\boldsymbol{y})e_{m'}(\boldsymbol{y}_{j'})\right)\left(\sum_{m''=1}^{\infty}\nu_{m''}^{-1}e_{m''}(\boldsymbol{y})e_{m''}(\boldsymbol{y}')\right)d\boldsymbol{y}d\boldsymbol{y}' \\
&= \sum_{m=1}^{\infty}\sum_{m'=1}^{\infty}\sum_{m''=1}^{\infty}\nu_m\nu_{m'}\nu_{m''}^{-1}e_m(\boldsymbol{y}_j)e_{m'}(\boldsymbol{y}_{j'})\left(\int_{\mathcal{Y}}e_m(\boldsymbol{y})e_{m''}(\boldsymbol{y})d\boldsymbol{y}\right)\left(\int_{\mathcal{Y}}e_{m'}(\boldsymbol{y}')e_{m''}(\boldsymbol{y}')d\boldsymbol{y}'\right) \\
&= \sum_{m=1}^{\infty}\sum_{m'=1}^{\infty}\sum_{m''=1}^{\infty}\nu_m\nu_{m'}\nu_{m''}^{-1}e_m(\boldsymbol{y}_j)e_{m'}(\boldsymbol{y}_{j'})\delta_{mm''}\delta_{m'm''} = \sum_{m=1}^{\infty}\nu_m e_m(\boldsymbol{y}_j)e_m(\boldsymbol{y}_{j'}) = k(\boldsymbol{y}_j,\boldsymbol{y}_{j'}),
\end{aligned}
$$

or equivalently,

$$
\underline{\mathcal{K}}\mathcal{K}^*\underline{\mathcal{K}}^\top = \boldsymbol{K}. \tag{S8}
$$

Substituting (S8) into (S6) yields the result,

$$
|\mathcal{H}| = |\mathcal{K}||\boldsymbol{W}^{-1} + \boldsymbol{K}|^{-1}|\boldsymbol{W}^{-1}| = |\mathcal{K}||\boldsymbol{I}_J + \boldsymbol{W}\boldsymbol{K}|^{-1}.
$$

## S4.2 Degenerate Approach

The equivalent kernel is constructed under the degenerate approach as follows:

$$
h(\boldsymbol{y},\boldsymbol{y}') = \boldsymbol{\phi}(\boldsymbol{y})^\top(\boldsymbol{I}_M + 2\boldsymbol{A})^{-1}\boldsymbol{\phi}(\boldsymbol{y}'),
$$

where

$$
k(\boldsymbol{y},\boldsymbol{y}') = \sum_{m=1}^{M}\phi_m(\boldsymbol{y})\phi_m(\boldsymbol{y}') = \boldsymbol{\phi}(\boldsymbol{y})^\top\boldsymbol{\phi}(\boldsymbol{y}'), \quad \boldsymbol{A} = \int_{\mathcal{Y}}\rho(\boldsymbol{y})\boldsymbol{\phi}(\boldsymbol{y})\boldsymbol{\phi}(\boldsymbol{y})^\top d\boldsymbol{y}.
$$

Mercer's theorem [5] states that the kernel function of finite rank $M$ has a diagonal representation such that

$$
k(\boldsymbol{y},\boldsymbol{y}') = \sum_{m=1}^{M}\nu_m e_m(\boldsymbol{y})e_m(\boldsymbol{y}') \quad \Leftrightarrow \quad \boldsymbol{\phi}(\boldsymbol{y}) = \boldsymbol{\Lambda}\boldsymbol{e}(\boldsymbol{y}), \quad \boldsymbol{\Lambda}_{mm'} = \sqrt{\nu_m}\delta_{mm'}, \tag{S9}
$$

where $\{e_m(\cdot)\}_m$ is an orthonormal basis comprising eigenfunctions of $\mathcal{K}$, and $\{\nu_m\}_m$ is the set of eigenvalues of $\mathcal{K}$. Using the relation (S9), we can rewrite the equivalent kernel in terms of the Mercer expansion,

$$
h(\boldsymbol{y},\boldsymbol{y}') = \boldsymbol{e}(\boldsymbol{y})^\top\boldsymbol{\Lambda}^\top(\boldsymbol{I}_M + 2\boldsymbol{A})^{-1}\boldsymbol{\Lambda}\boldsymbol{e}(\boldsymbol{y}') = (\boldsymbol{V}\boldsymbol{e}(\boldsymbol{y}))^\top\boldsymbol{\Xi}\,(\boldsymbol{V}\boldsymbol{e}(\boldsymbol{y}')), \tag{S10}
$$

where $\boldsymbol{\Xi}$ is a diagonal matrix whose diagonal entries are the eigenvalues of $\boldsymbol{\Lambda}^\top(\boldsymbol{I}_M + 2\boldsymbol{A})^{-1}\boldsymbol{\Lambda}$, and $\boldsymbol{V}$ is the modal matrix satisfying $\boldsymbol{V}^\top\boldsymbol{V} = \boldsymbol{I}_M$. Equation (S10) indicates that the eigenvalues of $\mathcal{H}$ is equal to that of $\boldsymbol{\Lambda}^\top(\boldsymbol{I}_M + 2\boldsymbol{A})^{-1}\boldsymbol{\Lambda}$, and thus the functional determinant of $\mathcal{H}$ is equal to the matrix determinant of $\boldsymbol{\Lambda}^\top(\boldsymbol{I}_M + 2\boldsymbol{A})^{-1}\boldsymbol{\Lambda}$,

$$
|\mathcal{H}| = |\boldsymbol{\Lambda}^\top(\boldsymbol{I}_M + 2\boldsymbol{A})^{-1}\boldsymbol{\Lambda}| = |\boldsymbol{\Lambda}^2||\boldsymbol{I}_M + 2\boldsymbol{A}|^{-1} = |\mathcal{K}||\boldsymbol{I}_M + 2\boldsymbol{A}|^{-1}.
$$

# S5 Experimental Settings and Additional Results

## S5.1 Model Configuration

### Augmented Permanental Process (APP)

Let the number of samples for quasi-Monte Carlo method be denoted by $J$, and the ranks of approximate kernel function for Random feature map [6] and Nyström approximation [8, 9] be denoted by $M_{\text{RFM}}$ and $M_{\text{NYS}}$, respectively. For all the experiments in Section 4, we used the following values:

$$
J = 2^{11} = 2048, \quad M_{\text{RFM}} = 100, \quad M_{\text{NYS}} = 500.
$$

Nyström approximation randomly selects $M_{\mathrm{NYS}}$ points from the training data points, and if $M_{\mathrm{NYS}}$ is larger than the number of the training data, $N$, then $M_{\mathrm{NYS}}$ is set as $N$.

We employed a popular gradient descent algorithm, *Adam* [4], to perform the minimization problem (see Section 2.2),

$$\{\hat{v}_n\}_{n=1}^N = \underset{\{v_n\}}{\arg\min}\, G, \qquad G \triangleq \frac{1}{N}\sum_{n=1}^N \left| 2\, v_n \sum_{n'=1}^N h(\boldsymbol{y}_n, \boldsymbol{y}_{n'}) v_{n'} - 1 \right|^2,$$

where the learning parameter ($lr$), the maximum number of iteration ($N_{\mathrm{ite}}$), and the stop condition were set as follows:

$$lr = 0.05(N/|\mathcal{T}|)^{-1/2}, \quad N_{\mathrm{ite}} = 500, \quad G < 10^{-5},$$

where $|\mathcal{T}|$ denotes the measure of observation domain $\mathcal{T}$. Here, $(N/|\mathcal{T}|)^{-1/2}$ represents the estimated value of $v_n = x(\boldsymbol{y}_n)^{-1}$ when the intensity is constant over domain $\mathcal{T}$.

We applied to the APPs a multiplicative Gaussian kernel, $k(\boldsymbol{y}, \boldsymbol{y}') = \theta_0 \prod_{d=1}^{D_y} e^{-(\theta_d(y_d - y'_d))^2}$, where the hyper-parameter $\theta = (\theta_0, \ldots, \theta_{D_y})$ was optimized for each data by maximizing the marginal likelihood through the 25-points grid search. As an initial estimate, we set a hyper-parameter as

$$\theta_0^* = \prod_{d=1}^{D_t} \mathrm{STD}\big[\{\tilde{x}_d^b\}_{b=1}^B\big]^{2/d}, \quad \theta_d^* = \mathrm{STD}\big[\{\boldsymbol{y}_d(\boldsymbol{t}_n)\}_{n=1}^N\big]^{-1} \quad \text{for } d = 1, \ldots, D_y, \tag{S11}$$

where STD represents the standard deviation, $\boldsymbol{y}_d(\boldsymbol{t})$ is the $d$-th element of $\boldsymbol{y}(\boldsymbol{t})$, $\tilde{x}_d^b$ is the square root of the intensity value estimated by a histogram density estimator with $B$ bins as,

$$\big(\tilde{x}_d^b\big)^2 = \big(\text{\# of data points } \in \big[\mathcal{T}_d^{\min} + \Delta(b-1)/B,\ \mathcal{T}_d^{\min} + \Delta b/B\big]\big)/\Delta, \quad \Delta = \mathcal{T}_d^{\max} - \mathcal{T}_d^{\min},$$

and $[\mathcal{T}_d^{\min}, \mathcal{T}_d^{\max}]$ is the boundary of $\mathcal{T}$ in each dimension $d$. $B$ was set as 10 in the experiments. Then we selected a set of five values for $\theta_0$ and $(\theta_1, \ldots, \theta_{D_y})$, respectively, as

$$\theta_0 = \{1/3, 1/2, 1, 2, 3\} \times \theta_0^*, \quad (\theta_1, \ldots, \theta_{D_y}) = \{1/3, 1/2, 1, 2, 3\} \times (\theta_1^*, \ldots, \theta_{D_y}^*),$$

and performed a grid search on the $5 \times 5 = 25$ hyper-parameter points. $\mathrm{APP}'_{\mathrm{RFM}}$ in Section 4.1 represents $\mathrm{APP}_{\mathrm{RFM}}$ with the initial hyper-parameter $(\theta_0^*, \theta_1^*, \ldots, \theta_{D_y}^*)$.

We implemented APPs by using TensorFlow-2.2.

**Kernel Intensity Estimator (KIE) and Local Likelihood Estimator (LLE)**

We implemented KIEs and LLEs through library `spatstat` in R [1]. We used `rhohat` for scenarios of one-dimensional covariate ($\mathcal{Y} \subset \mathbb{R}$), and `rho2hat` for scenarios of two-dimensional covariate ($\mathcal{Y} \subset \mathbb{R}^2$), where the variants of KIEs and LLEs were specified by arguments `method`, `smoother` and `bw`, while the other arguments were set as the default values.

**Integrated Nested Laplace Approximation (INLA)**

INLA discretizes the observation domain $\mathcal{T} \subset \mathcal{R}^{D_t}$ into $Q^{D_t}$ grid cells, and takes as input the observed number of points and the representative covariate value in each grid cell. For all the experiments in Section 4, we set $Q$ as 100. We implemented INLA through library `INLA` in R (https://www.r-inla.org/): The call in `INLA` to fit a model for one-dimensional covariate scenarios was

```
> formula = y ~ f(inla.group(cov, n = 100), model = "rw1",
                hyper = list(prec = list(param = prior)))
> result = inla(formula, data = data, family = "poisson",
                control.inla = list(strategy = "gaussian")),
```

where `cov` is the covariate values in grid cells, and `rw1` represents that the random walk model of order 1 is used as the prior process; The call in `INLA` to fit a model for two-dimensional covariate

scenarios was

$$> \texttt{formula} = \texttt{y} \sim \texttt{f}(\texttt{inla.group}(\texttt{cov1}, \texttt{n} = 100), \texttt{model} = \text{``rw1''},$$
$$\texttt{hyper} = \texttt{list}(\texttt{prec} = \texttt{list}(\texttt{param} = \texttt{prior})))$$
$$+ \texttt{f}(\texttt{inla.group}(\texttt{cov2}, \texttt{n} = 100), \texttt{model} = \text{``rw1''},$$
$$\texttt{hyper} = \texttt{list}(\texttt{prec} = \texttt{list}(\texttt{param} = \texttt{prior})))$$
$$> \texttt{result} = \texttt{inla}(\texttt{formula}, \texttt{data} = \texttt{data}, \texttt{family} = \text{``poisson''},$$
$$\texttt{control.inla} = \texttt{list}(\texttt{strategy} = \text{``gaussian''})),$$

where `cov1` and `cov2` represent the first and the second elements of 2D covariate values in grid cells, respectively. `prior` represents the hyper-parameter of prior process, and was optimized by maximizing the marginal likelihood through grid search: Letting the hyper-parameter be denoted by $\boldsymbol{\xi} = (\xi_1, \xi_2)$, we selected a set of four values for each element of $\boldsymbol{\xi}$ as

$$\xi_1 = \{0.1, 1, 10, 100\}, \quad \xi_2 = \{0.00005, 0.0001, 0.001, 0.01\}.$$

and performed a grid search on the $4 \times 4 = 16$ hyper-parameter points.

## S5.2 Details of Covariate Map

In synthetic data experiments, we created a covariate map $d_R(\boldsymbol{t})$, which was defined as the shortest distance from a given location $\boldsymbol{t} \in \mathcal{T}$ to the set of lines arranged in the shape of the letter "R", and the covariate map was represented by a $100 \times 100$ pixel grid. KIEs/LLEs used the pixel image of covariate map as argument `covariate`, while APPs and INLA constructed a continuous covariate map based on the pixel image by using a linear interpolation method.

In `copper`, we created a covariate map as the shortest distance from a given location $\boldsymbol{t}$ to the set of line segments representing faults, and represented the covariate map by a $512 \times 512$ pixel grid. In `bei` and `clmfires`, the covariate maps are provided as a $101 \times 201$ pixel grid and a $200 \times 200$ pixel grid, respectively. The pixel images of covariate map were used with the same procedure as in synthetic data.

## S5.3 Performance Metrics

In synthetic data experiments, the predictive performance was evaluated based on the integrated *$\rho$-quantile loss* [7], defined as

$$l_\rho \triangleq \int_0^{1.5} 2\big(g(y) - \hat{g}(y)\big)\big(\rho \mathrm{I}_{g(y) > \hat{g}(y)} - (1 - \rho)\mathrm{I}_{g(y) \leq \hat{g}(y)}\big) dy,$$

where I, $\hat{g}(y)$, and $g(y)$ denote the indicator, the predicted $\rho$-quantile of the intensity function on covariate domain, and the true one, respectively. The integral was computed via 2000-points numerical integration.

In real-world data experiments, the predictive performances were evaluated based on the negative test log likelihood of point patterns ($ll_{\text{test}}$) and the negative test likelihood of counts ($cl_{\text{test}}$): $ll_{\text{test}}$ was computed as

$$ll_{\text{test}} = - \sum_{\boldsymbol{t} \in \mathcal{D}_{\text{test}}} \log \hat{\lambda}(\boldsymbol{t}) + \int_{\mathcal{T}} \hat{\lambda}(\boldsymbol{t}) d\boldsymbol{t},$$

where $\hat{\lambda}(\boldsymbol{t})$ is the estimated intensity function (e.g. $\hat{\lambda}(\boldsymbol{t}) = \hat{x}^2(\boldsymbol{y}(\boldsymbol{t}))$ in APPs); the observation domain $\mathcal{T} \subset \mathcal{R}^2$ was discretized into $5 \times 5$ grid cells, and $cl_{\text{test}}$ was computed as

$$cl_{\text{test}} = \sum_{c \in \mathcal{C}_{\text{grid}}} (\Lambda_c - n_c \log \Lambda_c - \log(n_c!)), \quad \Lambda_c = \int_{\mathcal{T}^c} \hat{\lambda}(\boldsymbol{t}) d\boldsymbol{t},$$

where $\mathcal{C}_{\text{grid}}$ is the set of 25 grid cells, $n_c$ is the number of test event points observed in grid cell $c \in \mathcal{C}_{\text{grid}}$, and $\mathcal{T}^c$ is the domain of grid cell $c$. The 2D integral was evaluated via $500 \times 500$ points numerical integration.

Table S1: Results on two types of synthetic data across 100 trials with standard errors in brackets. The underlines represent the best predictive performances on each metric, and the performances not significantly ($p < 10^{-2}$) different from the best one under the Mann-Whitney U test with Holm method [2] are shown in bold. $cpu$ is the CPU times in second, and $\tilde{N}$ is the average data size.

| $g_1(d_R)$ | $\alpha = 0.2$ ($\tilde{N} = 232$) | | | | $\alpha = 0.5$ ($\tilde{N} = 586$) | | | | $\alpha = 1.0$ ($\tilde{N} = 1180$) | | | |
|---|---|---|---|---|---|---|---|---|---|---|---|---|
| | $l_{.025}$ | $l_{.5}$ | $l_{.975}$ | $cpu$ | $l_{.025}$ | $l_{.5}$ | $l_{.975}$ | $cpu$ | $l_{.025}$ | $l_{.5}$ | $l_{.975}$ | $cpu$ |
| KIEra$_{SI}$ | 0.17 (0.06) | 1.48 (0.35) | 0.36 (0.15) | 0.05 (0.00) | 0.40 (0.32) | 2.89 (0.63) | 0.59 (0.21) | 0.06 (0.00) | 1.14 (0.74) | 5.17 (1.12) | 0.86 (0.26) | 0.06 (0.00) |
| KIEra$_{SJ}$ | 0.19 (0.06) | 1.76 (0.34) | 0.66 (0.30) | 0.05 (0.00) | 0.37 (0.20) | 3.25 (0.61) | 1.18 (0.50) | 0.06 (0.00) | 0.54 (0.24) | 4.90 (0.81) | 1.58 (0.58) | 0.06 (0.00) |
| KIEre$_{SI}$ | 0.19 (0.09) | 1.71 (0.35) | 0.56 (0.18) | 0.05 (0.00) | 0.59 (0.42) | 3.45 (0.65) | 1.05 (0.30) | 0.06 (0.00) | 1.90 (0.92) | 6.18 (1.11) | 1.62 (0.33) | 0.06 (0.00) |
| KIEre$_{SJ}$ | 0.19 (0.04) | 1.82 (0.32) | 0.90 (0.26) | 0.05 (0.01) | 0.36 (0.13) | 3.41 (0.55) | 1.80 (0.43) | 0.06 (0.00) | 0.56 (0.21) | 5.39 (0.79) | 2.89 (0.47) | 0.06 (0.00) |
| KIEtr$_{SI}$ | 0.16 (0.10) | 1.26 (0.41) | 0.22 (0.10) | 0.05 (0.00) | 0.61 (0.60) | 2.79 (0.70) | 0.40 (0.14) | 0.06 (0.00) | 2.15 (1.04) | 5.13 (1.03) | 0.62 (0.18) | 0.06 (0.00) |
| KIEtr$_{SJ}$ | 0.15 (0.05) | **1.16** (0.35) | 0.21 (0.10) | 0.05 (0.00) | 0.29 (0.20) | 2.26 (0.52) | 0.36 (0.10) | 0.06 (0.00) | 0.43 (0.26) | 3.45 (0.84) | 0.51 (0.12) | 0.06 (0.00) |
| LLEra | 0.17 (0.09) | 1.41 (0.39) | 0.23 (0.05) | 0.07 (0.00) | 0.43 (0.34) | 2.91 (0.80) | 0.57 (0.29) | 0.08 (0.00) | 1.11 (0.85) | 5.05 (1.11) | 0.92 (0.45) | 0.08 (0.00) |
| LLEre | 0.37 (0.01) | 1.45 (0.40) | 1.71 (0.05) | 0.07 (0.00) | 0.71 (0.03) | 2.87 (0.77) | 1.95 (0.07) | 0.08 (0.00) | 1.10 (0.05) | 4.87 (1.05) | 2.38 (0.10) | 0.08 (0.00) |
| LLEtr | 0.16 (0.07) | 1.47 (0.39) | 0.29 (0.11) | 0.07 (0.00) | 0.37 (0.27) | 3.08 (0.76) | 1.00 (0.44) | 0.08 (0.00) | 0.91 (0.71) | 5.26 (0.97) | 2.13 (0.53) | 0.08 (0.00) |
| INLA | 0.27 (0.03) | 1.31 (0.30) | 0.62 (0.13) | 101 (6.01) | 0.54 (0.06) | 2.54 (0.51) | 1.00 (0.14) | 108 (5.19) | 0.89 (0.07) | 3.71 (0.70) | 1.51 (0.20) | 110 (6.10) |
| APP$_{NAI}$ | **0.14** (0.06) | **1.08** (0.32) | **0.18** (0.09) | 9.30 (0.52) | **0.24** (0.10) | **1.93** (0.63) | **0.34** (0.22) | 14.1 (0.80) | **0.38** (0.26) | **2.74** (0.78) | **0.45** (0.33) | 25.5 (1.05) |
| APP$_{NYS}$ | **0.14** (0.05) | **1.08** (0.31) | **0.18** (0.09) | 0.61 (0.09) | **0.24** (0.10) | **1.93** (0.62) | **0.34** (0.22) | 2.62 (0.19) | **0.38** (0.27) | **2.72** (0.78) | **0.44** (0.33) | 2.81 (0.35) |
| APP$_{RFM}$ | **0.13** (0.07) | **1.06** (0.33) | **0.18** (0.13) | 2.42 (0.17) | **0.24** (0.10) | **1.90** (0.60) | 0.34 (0.24) | 2.48 (0.07) | **0.37** (0.27) | **2.65** (0.75) | **0.43** (0.34) | 2.56 (0.05) |
| APP$'_{RFM}$ | 0.17 (0.04) | 1.38 (0.32) | 0.24 (0.07) | 0.30 (0.01) | 0.31 (0.12) | 2.48 (0.64) | 0.45 (0.23) | 0.32 (0.01) | 0.49 (0.24) | 3.66 (0.82) | 0.60 (0.29) | 0.35 (0.01) |

| $g_2(d_R)$ | $\alpha = 0.2$ ($\tilde{N} = 322$) | | | | $\alpha = 0.5$ ($\tilde{N} = 801$) | | | | $\alpha = 1.0$ ($\tilde{N} = 1610$) | | | |
|---|---|---|---|---|---|---|---|---|---|---|---|---|
| | $l_{.025}$ | $l_{.5}$ | $l_{.975}$ | $cpu$ | $l_{.025}$ | $l_{.5}$ | $l_{.975}$ | $cpu$ | $l_{.025}$ | $l_{.5}$ | $l_{.975}$ | $cpu$ |
| KIEra$_{SI}$ | 0.21 (0.06) | 1.65 (0.35) | 0.31 (0.15) | 0.05 (0.00) | 0.60 (0.32) | 3.44 (0.63) | 0.53 (0.21) | 0.06 (0.00) | 2.25 (0.74) | 6.41 (1.12) | 0.83 (0.26) | 0.06 (0.00) |
| KIEra$_{SJ}$ | 0.21 (0.06) | 1.73 (0.34) | 0.37 (0.30) | 0.05 (0.00) | 0.40 (0.20) | 3.17 (0.61) | 0.65 (0.50) | 0.06 (0.00) | 0.66 (0.24) | 4.94 (0.81) | 0.90 (0.58) | 0.06 (0.00) |
| KIEre$_{SI}$ | 0.25 (0.09) | 1.79 (0.35) | 0.38 (0.18) | 0.05 (0.00) | 0.78 (0.42) | 3.82 (0.65) | 0.75 (0.30) | 0.06 (0.00) | 2.70 (0.92) | 7.01 (1.11) | 1.24 (0.33) | 0.06 (0.00) |
| KIEre$_{SJ}$ | 0.21 (0.04) | 1.78 (0.32) | 0.60 (0.26) | 0.05 (0.01) | 0.39 (0.13) | 3.41 (0.55) | 1.29 (0.43) | 0.06 (0.00) | 0.65 (0.21) | 5.55 (0.79) | 2.21 (0.47) | 0.06 (0.00) |
| KIEtr$_{SI}$ | 0.32 (0.10) | 1.79 (0.41) | 0.26 (0.10) | 0.05 (0.00) | 1.19 (0.60) | 3.74 (0.70) | 0.48 (0.14) | 0.06 (0.00) | 3.80 (1.04) | 6.92 (1.03) | 0.77 (0.18) | 0.06 (0.00) |
| KIEtr$_{SJ}$ | 0.18 (0.05) | 1.47 (0.35) | 0.24 (0.10) | 0.06 (0.00) | 0.38 (0.20) | 2.76 (0.52) | 0.40 (0.10) | 0.06 (0.00) | 0.83 (0.26) | 4.77 (0.84) | 0.64 (0.12) | 0.06 (0.00) |
| LLEra | 0.24 (0.09) | 1.67 (0.39) | 0.28 (0.05) | 0.07 (0.00) | 0.74 (0.34) | 3.47 (0.80) | 0.54 (0.29) | 0.08 (0.00) | 1.95 (0.85) | 6.17 (1.11) | 1.02 (0.45) | 0.08 (0.00) |
| LLEre | 0.44 (0.01) | 1.68 (0.40) | 1.74 (0.05) | 0.07 (0.00) | 0.83 (0.03) | 3.47 (0.77) | 2.16 (0.07) | 0.08 (0.00) | 1.25 (0.05) | 6.16 (1.05) | 2.75 (0.10) | 0.08 (0.00) |
| LLEtr | 0.22 (0.07) | 1.69 (0.39) | 0.31 (0.11) | 0.07 (0.00) | 0.59 (0.27) | 3.61 (0.76) | 0.95 (0.44) | 0.08 (0.00) | 1.50 (0.71) | 6.59 (0.97) | 2.24 (0.53) | 0.08 (0.00) |
| INLA | 0.45 (0.03) | 1.73 (0.30) | 1.34 (0.13) | 112 (6.01) | 0.96 (0.06) | 3.31 (0.51) | 2.22 (0.14) | 115 (5.19) | 1.68 (0.07) | 5.04 (0.70) | 3.45 (0.20) | 116 (6.10) |
| APP$_{NAI}$ | **0.15** (0.06) | **1.06** (0.32) | **0.17** (0.09) | 10.1 (0.52) | **0.25** (0.10) | **1.87** (0.63) | **0.27** (0.22) | 17.7 (0.80) | **0.40** (0.26) | **2.75** (0.78) | **0.44** (0.33) | 40.1 (1.05) |
| APP$_{NYS}$ | **0.15** (0.05) | **1.08** (0.31) | **0.18** (0.09) | 1.02 (0.09) | **0.25** (0.10) | **1.88** (0.62) | **0.27** (0.22) | 2.72 (0.19) | **0.39** (0.27) | **2.79** (0.78) | **0.44** (0.33) | 3.15 (0.35) |
| APP$_{RFM}$ | **0.14** (0.07) | **0.99** (0.33) | **0.17** (0.13) | 2.40 (0.17) | **0.22** (0.10) | **1.76** (0.60) | **0.26** (0.24) | 2.57 (0.07) | **0.36** (0.27) | **2.53** (0.75) | **0.40** (0.34) | 2.78 (0.05) |
| APP$'_{RFM}$ | 0.20 (0.04) | 1.57 (0.32) | 0.28 (0.07) | 0.30 (0.01) | 0.37 (0.12) | 2.76 (0.64) | 0.46 (0.23) | 0.34 (0.01) | 0.57 (0.24) | 4.06 (0.82) | 0.65 (0.29) | 0.40 (0.01) |

Table S2: Results on real-world data across 10 trials. Notations follow Table S1.

| | copper | | | bei | | | clmfires | | |
|---|---|---|---|---|---|---|---|---|---|
| | $l_{\text{test}}$ | $cl_{\text{test}}$ | $cpu$ | $l_{\text{test}}$ | $cl_{\text{test}}$ | $cpu$ | $l_{\text{test}}$ | $cl_{\text{test}}$ | $cpu$ |
| $\text{KIEra}_{\text{SI}}$ | **5.65** (0.37) | **1.64** (0.09) | 0.32 (0.01) | 2.92 (0.00) | 1.31 (0.07) | 0.26 (0.00) | 2.39 (0.00) | 1.30 (0.08) | 0.24 (0.01) |
| $\text{KIEra}_{\text{SJ}}$ | **5.65** (0.38) | **1.64** (0.09) | 0.33 (0.01) | 2.92 (0.00) | 1.31 (0.07) | 0.25 (0.00) | 2.39 (0.00) | 1.30 (0.08) | 0.24 (0.01) |
| $\text{KIEre}_{\text{SI}}$ | **5.65** (0.37) | **1.64** (0.09) | 0.33 (0.01) | 2.92 (0.00) | 1.31 (0.07) | 0.25 (0.00) | 2.40 (0.00) | 1.38 (0.08) | 0.24 (0.01) |
| $\text{KIEre}_{\text{SJ}}$ | **5.65** (0.38) | **1.64** (0.09) | 0.34 (0.02) | 2.92 (0.00) | 1.31 (0.07) | 0.25 (0.01) | 2.40 (0.00) | 1.38 (0.08) | 0.24 (0.00) |
| INLA | 5.64 (0.34) | 1.66 (0.10) | 82.9 (3.00) | 2.90 (0.01) | 1.13 (0.06) | 131 (3.37) | 2.39 (0.01) | **1.18** (0.06) | 129 (4.61) |
| $\text{APP}_{\text{NAI}}$ | **5.65** (0.37) | **1.64** (0.09) | 8.21 (0.50) | 2.96 (0.03) | 1.51 (0.29) | 200 (4.67) | 2.42 (0.03) | 1.38 (0.10) | 304 (8.97) |
| $\text{APP}_{\text{NYS}}$ | **5.65** (0.37) | **1.64** (0.09) | 0.60 (0.06) | **2.87** (0.01) | **1.01** (0.05) | 15.6 (0.41) | **2.36** (0.01) | **1.15** (0.07) | 17.9 (1.06) |
| $\text{APP}_{\text{RFM}}$ | **5.65** (0.37) | **1.64** (0.09) | 2.12 (0.10) | **2.88** (0.01) | **1.02** (0.05) | 4.64 (0.21) | **2.36** (0.01) | **1.16** (0.07) | 4.28 (0.22) |

## S5.4 Full Results

Table S1 and Table S2 display the predictive performances of all compared methods on synthetic and real-world data, respectively.

## S6 Experiment on Larger Synthetic Data Set

We created data sets according to the scenario $\lambda_1(t) = g_1(d_R(t)) = \alpha \exp(5 - 3d_R(t))$ (see Section 4.1), in which the size of each data sets was controlled by the coefficient $\alpha$ from $\alpha = 0.5$ to $\alpha = 50$, resulting in the training data sets containing from 589 to 23,653 data points on average. The computational complexity of INLA depends not on the data size but on the size of domain discretization, and thus we compared $\text{APP}_{\text{RFM}}$ with INLAs of $10 \times 10$ ($\text{INLA}_{10}$), $50 \times 50$ ($\text{INLA}_{50}$), $100 \times 100$ ($\text{INLA}_{100}$), and $150 \times 150$ ($\text{INLA}_{150}$) domain discretization.

Table S3 and Figure S1 display the predictive performances and the CPU times as function of the data size, showing that our $\text{APP}_{\text{RFM}}$ achieved better predictive performances than INLAs, while $\text{APP}_{\text{RFM}}$ was performed substantially faster than INLAs for at least up to tens of thousands of data points. The performance gaps were not marginal (e.g., the improvements of the integrated absolute error $l_{.5}$ were $18\% \sim 38\%$). Figure S3 displays the predictive performances multiplied by the CPU times (the lower, the better), which represent the predictive performances penalized by its execution times, that is, the ratios of predictive performance to speed. Figure S3 also shows that our $\text{APP}_{\text{RFM}}$ is beneficial against INLAs.

## S7 Experiment on Synthetic Data of 2D Covariate

We created 2D data sets with 2D covariate ($\mathcal{T} \subset \mathbb{R}^2$, $\mathcal{Y} \subset \mathbb{R}^2$) generated from the following intensity function: $\lambda(t) = 0.01\alpha \exp(5 - 3d_R(t)) * \exp(5 - 4d_Z(t)^2)$, which has 20 trial sequences. Here $d_R(t)$ and $d_Z(t)$ denote the shortest distances from a given location $t$ to the sets of lines arranged in the shapes of the letters "R" and "Z", respectively. The covariate map is given as $y(t) = (d_R(t), d_Z(t))$. The coefficient $\alpha$ was set as 0.5 and 1.0. The predictive performance was evaluated based on the integrated $\rho$-*quantile loss* [7], defined as $l_\rho \triangleq \int_{\mathcal{Y}} 2(g(y) - \hat{g}(y))(\rho I_{g(y) > \hat{g}(y)} - (1-\rho)I_{g(y) \leq \hat{g}(y)}) dy$, where I, $\hat{g}(y)$, and $g(y)$ denote the indicator, the predicted $\rho$-quantile of the intensity function on covariate domain, and the true one, respectively. Here, we adopted $l_{.025}$, $l_{.5}$ (integrated absolute error) and $l_{.975}$. Table S4 shows the result.

Table S3: Results on larger synthetic data sets across 20 trials with standard errors in brackets. $cpu$ is the CPU times in second, and $\tilde{N}$ is the average data size for each data set.

| | $(\alpha, \tilde{N}) = (0.5, 589)$ | | | | $(\alpha, \tilde{N}) = (1.0, 1169)$ | | | | $(\alpha, \tilde{N}) = (5.0, 5921)$ | | | |
| | $l_{.025}$ | $l_{.5}$ | $l_{.975}$ | $cpu$ | $l_{.025}$ | $l_{.5}$ | $l_{.975}$ | $cpu$ | $l_{.025}$ | $l_{.5}$ | $l_{.975}$ | $cpu$ |
|---|---|---|---|---|---|---|---|---|---|---|---|---|
| INLA$_{10}$ | 0.41 | 4.25 | 0.76 | 80.3 | 0.66 | 6.33 | 1.15 | 84.1 | 5.84 | 30.8 | 11.5 | 88.9 |
| | (0.11) | (0.72) | (0.18) | (6.32) | (0.14) | (1.11) | (0.30) | (8.61) | (1.22) | (2.70) | (2.38) | (11.1) |
| INLA$_{50}$ | 0.48 | 2.66 | 0.91 | 87.9 | 0.81 | 3.96 | 1.25 | 90.4 | 2.39 | 10.5 | 3.34 | 95.8 |
| | (0.06) | (0.58) | (0.13) | (10.9) | (0.07) | (0.83) | (0.12) | (7.36) | (0.16) | (2.16) | (0.23) | (12.0) |
| INLA$_{100}$ | 0.52 | 2.68 | 1.05 | 114 | 0.89 | 3.79 | 1.42 | 118 | 2.76 | 10.8 | 3.92 | 114 |
| | (0.06) | (0.64) | (0.15) | (9.53) | (0.07) | (0.95) | (0.14) | (9.10) | (0.18) | (1.88) | (0.33) | (11.0) |
| INLA$_{150}$ | 0.53 | 2.64 | 1.06 | 165 | 0.91 | 3.85 | 1.49 | 166 | 2.85 | 10.4 | 4.02 | 151 |
| | (0.06) | (0.60) | (0.15) | (14.0) | (0.07) | (0.99) | (0.18) | (12.3) | (0.18) | (1.77) | (0.33) | (11.6) |
| APP$_{RFM}$ | 0.24 | 2.16 | 0.36 | 2.43 | 0.36 | 2.60 | 0.45 | 2.58 | 1.10 | 6.60 | 1.20 | 4.58 |
| | (0.06) | (0.61) | (0.32) | (0.08) | (0.17) | (0.81) | (0.41) | (0.04) | (0.83) | (2.06) | (0.74) | (0.24) |

| | $(\alpha, \tilde{N}) = (10, 11808)$ | | | | $(\alpha, \tilde{N}) = (20, 23653)$ | | | |
| | $l_{.025}$ | $l_{.5}$ | $l_{.975}$ | $cpu$ | $l_{.025}$ | $l_{.5}$ | $l_{.975}$ | $cpu$ |
|---|---|---|---|---|---|---|---|---|
| INLA$_{10}$ | 17.1 | 59.4 | 27.9 | 97.9 | 50.8 | 119 | 64.0 | 97.5 |
| | (3.07) | (4.65) | (4.85) | (13.5) | (5.88) | (6.56) | (9.01) | (13.5) |
| INLA$_{50}$ | 3.89 | 17.6 | 4.99 | 102 | 6.20 | 28.9 | 7.81 | 101 |
| | (0.26) | (2.49) | (0.36) | (10.9) | (0.32) | (4.47) | (0.58) | (12.0) |
| INLA$_{100}$ | 4.54 | 16.8 | 5.88 | 124 | 7.48 | 28.5 | 9.38 | 121 |
| | (0.28) | (2.04) | (0.40) | (12.7) | (0.35) | (3.78) | (0.74) | (12.2) |
| INLA$_{150}$ | 4.72 | 17.2 | 6.17 | 160 | 7.73 | 27.4 | 9.81 | 153 |
| | (0.31) | (2.89) | (0.43) | (13.0) | (0.30) | (4.83) | (0.75) | (11.6) |
| APP$_{RFM}$ | 1.21 | 10.6 | 2.13 | 8.73 | 4.98 | 20.6 | 4.89 | 18.4 |
| | (0.35) | (2.65) | (1.47) | (1.01) | (5.46) | (10.0) | (5.25) | (2.67) |

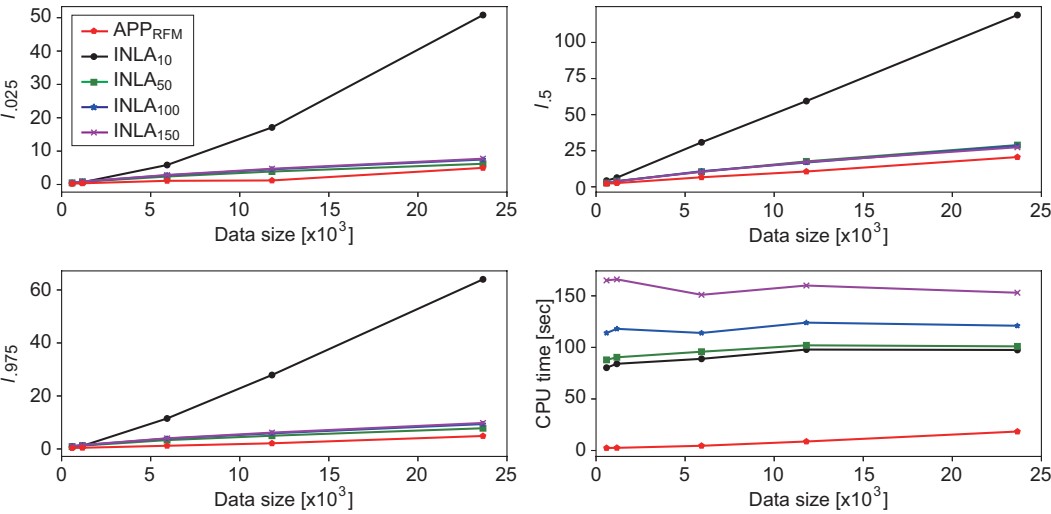

Figure S1: Results on larger synthetic data sets across 20 trials. The predictive performances and the CPU times as function of the data size: the lower, the better.

## S8   Potential Negative Societal Impacts

Although our model itself does not contain either any ethical problems or negative societal impacts, it could predict fine spatio-temporal patterns of people's behaviors, which might harm their privacy in some cases, and thus great care should be taken to protect personal information.

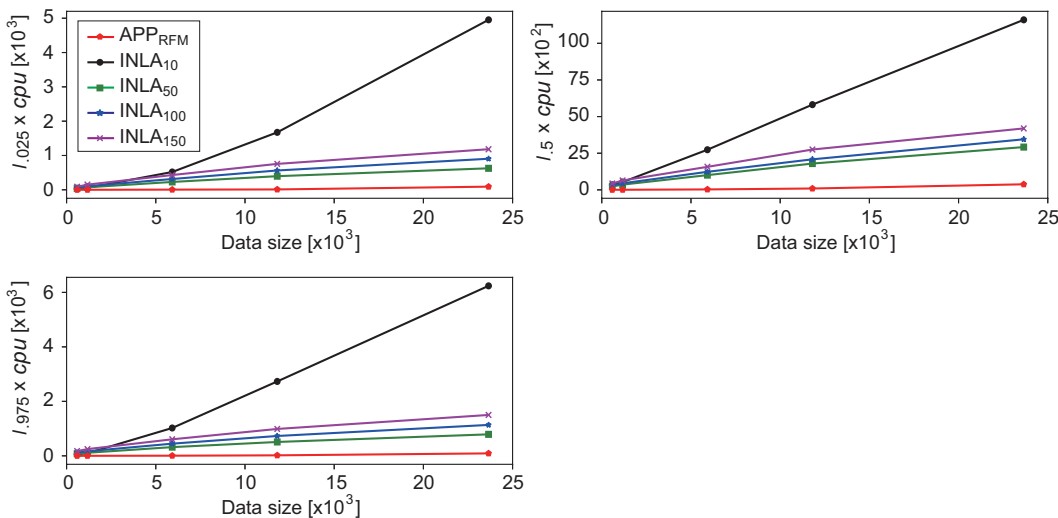

Figure S2: Results on larger synthetic data sets across 20 trials. The predictive performances penalized by the CPU times as function of the data size: the lower, the better.

Table S4: Results on synthetic data with 2D covariate across 20 trials. $cpu$ is the CPU time in second, and the underlines represent the best predictive performances on each metric. Notations follow Table S1.

| $\lambda_1(y)$ | $\alpha = 0.5$ | | | | $\alpha = 1.0$ | | | |
|---|---|---|---|---|---|---|---|---|
| | $l_{.025}$ | $l_{.5}$ | $l_{.975}$ | $cpu$ | $l_{.025}$ | $l_{.5}$ | $l_{.975}$ | $cpu$ |
| KIEra$_{SI}$ | 19.7 | 11.0 | 2.31 | 0.17 | 40.4 | 22.6 | 4.73 | 0.18 |
| | (2.42) | (1.20) | (0.29) | (0.01) | (3.43) | (1.68) | (0.37) | (0.01) |
| KIEra$_{SJ}$ | 19.7 | 11.0 | 2.31 | 0.16 | 40.4 | 22.6 | 4.73 | 0.19 |
| | (2.41) | (1.20) | (0.29) | (0.01) | (3.43) | (1.68) | (0.37) | (0.02) |
| KIEre$_{SI}$ | 2.60 | **3.57** | 4.54 | 0.17 | 5.22 | **7.01** | 8.80 | 0.19 |
| | (0.83) | (0.17) | (0.61) | (0.01) | (1.23) | (0.30) | (0.82) | (0.01) |
| KIEre$_{SJ}$ | 2.60 | **3.57** | 4.54 | 0.16 | 5.22 | **7.01** | 8.80 | 0.18 |
| | (0.83) | (0.17) | (0.61) | (0.01) | (1.22) | (0.30) | (0.82) | (0.01) |
| INLA | 0.78 | **4.13** | 7.75 | 143 | 1.59 | **7.44** | 9.81 | 145 |
| | (0.06) | (0.86) | (1.60) | (4.63) | (0.28) | (1.76) | (1.22) | (6.50) |
| APP$_{NYS}$ | **0.39** | 3.64 | **0.87** | 7.19 | **0.63** | **5.94** | **1.49** | 8.87 |
| | (0.08) | (0.71) | (0.15) | (0.68) | (0.11) | (1.04) | (0.21) | (0.58) |
| APP$_{RFM}$ | **0.38** | 4.70 | 1.79 | 2.46 | **0.59** | 7.69 | 3.04 | 2.95 |
| | (0.09) | (0.86) | (0.26) | (0.12) | (0.08) | (1.32) | (0.44) | (0.26) |