# OpenReview forum: "Fast Bayesian Estimation of Point Process Intensity as Function of Covariates"
_NeurIPS.cc/2022/Conference — NeurIPS 2022 Accept_

### Official Review · Reviewer_wjWf · 2022-06-27

**Rating:** 5
**Confidence:** 4
**Soundness:** 3 good
**Presentation:** 3 good
**Contribution:** 2 fair

**Summary:**

This paper proposed an augmented permanental process whose intensity is modeled by a Gaussian process on covariate space, and derive a efficient Bayesian inference method without the need of discretizing the domain. The proposed algorithm is based on path integration and Laplace approximation, which facilitates the computation of predictive distribution (uncertainty) and marginal distribution (hyper-parameter). Also the paper designed different methods, including naive methods, Nyström approximation and random features, to construct the equivalent kernel.

**Questions:**

I suggest to provide more experimental comparison results with SOTA baseline models, e.g., sigmoidal Cox process [2], permanental process [3]. Provide more analysis expeirmentally or theoretically on the ' Nodal Lines' problem which is a common issue in permanental processes.


[2] Efficient Bayesian Inference of Sigmoidal Gaussian Cox Processes. JMLR 2018.
[3] Large-Scale Cox Process Inference using Variational Fourier Features. ICML 2018.

**Limitations:**

The work does not have potential negative societal impact.
The authors did not mention the limitations of their work if I did not miss any important points.

**Strengths And Weaknesses:**

strengths:
The paper is writting very well and easy to follow. The methodology is solid and seems correct to me but I did not check all the derivations carefully.

weaknesses:
However, I have two concerns about the current version, which makes me hasitate on the rating of this submission.
The first is the experimental comparision with baseline models is rather weak. The proposed methods (NAI/NYS/RFM) are only compared with very old works. For example, KIE and INLA are published in 2012. I did not find any SOTA baselines in this work, which cannot convince me of the merits of this work.
The second is the contribution of this work is rather incremental. The key components of the proposed inference method all come from previous works, e.g., the MAP estimate using path integral is from [1], let alone the Laplace approximation of posterior and Nyström or random features approximation of kernel.

[1] Fast Bayesian inference for Gaussian Cox processes via path integral formulation. Neurips 2021.

---

> ### Author Response · Authors · 2022-08-02
> **Responses to Reviewer wjWf**
>
> We would like to thank the reviewer for the careful reading and the constructive comments. Below we provide a detailed response to each of the comments. We believe that we can clear up the two major concerns raised。
>
>
>
> **the experimental comparison with baseline models is rather weak. The proposed methods  (NAI/NYS/RFM) are only compared with very old works. For example, KIE and INLA are published in 2012. ... I suggest to provide more experimental comparison results with SOTA baseline models, e.g.,  sigmoidal Cox process [2], permanental process [3].**
>
> It is true that KIE and INLA seem to be old methods published in 2008-2012, but to the best of our knowledge, there has been no feasible methods proposed to address the problem "inferring intensity as function of covariates" other than KIE and INLA (an exception is a neural network-based model, see Related Work). Sigmoidal Cox process [2] and permanental process [3], suggested by the reviewer, are the methods for another problem "inferring intensity as function of every point in observation domain", and they cannot be compared directly with our proposed method. Surprisingly, the problem of "inferring intensity as function of covariates" has been overlooked in the recent machine learning community despite its great practical importance, which is the reason why we started this study. Therefore, KIE and INLA are SOTA baselines for the problem we tackled without a doubt.
>
> We recognize from the reviewer's comment that the problem we tackled is a little confusing (also see our response to reviewer V1pm). Thus we promise to add in Introduction the discussion above. We hope that the reviewer's concern is now fully cleared.
>
>
>
> **the contribution of this work is rather incremental. The key components of the proposed inference method all come from previous works,**
>
> The main contribution of this paper is a finding that the representer theorem, one of the most desirable mathematical property for machine learning problems, holds for the augmented permanental process. The finding brings a modern SOTA algorithm to the practically important problem "inferring point process intensity as function of covariates", which has been overlooked in the recent machine learning community. Although the key derivation of algorithms are based on the methodology proposed in [Hideaki Kim 2021], we believe that the result makes a non-incremental impact on the Bayesian inference community for point processes. Also, our finding makes a non-trivial contribution to the reproducing kernel Hilbert space community because it shows a novel and practically important likelihood function (Eq. (2)) for which the representer theorem holds under the regularization term of the squared Hilbert space norm.
>
> We will add the explanation of our contribution in the main text. We would be pleased if our explanation is satisfactory to the reviewer.
>
>
>
> **The authors did not mention the limitations of their work if I did not miss any important points.**
>
> We will add a detailed discussion about our method's limitations. Please see our 3rd response to reviewer oHGh.

---

### Official Review · Reviewer_oHGh · 2022-07-04

**Rating:** 5
**Confidence:** 3
**Soundness:** 3 good
**Presentation:** 3 good
**Contribution:** 2 fair

**Summary:**

This paper studies a fast Bayesian inference of point process intensity as a function of covariate.
- Using the path integral formulation of Gaussian processes, the maximum a posteriori (MAP) estimate can be obtained by minimizing the action integral, which results in an optimization problem with O(N^2) time complexity for each evaluation.
- In particular, as the quadratic link function is used in this context, the Nystroem method, and degenerate kernel method are employed to solve integral equations.

**Questions:**

Can you provide the running time comparisons of your method against the conventional Bayesian method, such as INLA, for the final experiments?

**Ethics Review Area:**

["I don’t know"]

**Limitations:**

The authors claim that the proposed method may harm people's privacy by predicting their fine temporal patterns.
I expect the authors to present more in-depth analysis about the limitations of the method.

**Strengths And Weaknesses:**

Originality:
-The novelty of this paper lies in the nonlinear dependence of the latent intensity on covariates, \lambda( K( y(t) ) ), compared to the previous work [Hideaki Kim 2021], in which the latent intensity is given by \lambda( K( t ) ).

- In particular, this paper employs quadratic link function, and thus enables to obtain a linear integral equation, which is first solved using Nystroem method, and degenerate kernel by this paper.

- I consider that most results in deriving MAP estimate, marginal likelihood and predictive distribution in Sec 2.2 and 2.3 are a bit incremental, given the results by [Hideaki Kim 2021].

Quality:
The work appears to be sound.

Clarity:
Overall, the methodology is well documented, and the main derivations are easy to follow.

Significance:
The work comes up a faster Bayesian inference method for covariates-dependent point processes, compared to previous MCMC and VB methods using domain discretizations. Thus, I would think it will make some impacts on the Bayesian inference community for point processes although the novelty is not that strong.

---

> ### Author Response · Authors · 2022-08-02
> **Responses to Reviewer oHGh**
>
> We would like to thank the reviewer for the careful reading and the constructive comments. Below we provide a detailed response to each of the comments.
>
>
>
> **I would think it will make some impacts on the Bayesian inference community for point  processes although the novelty is not that strong.**
>
> The main contribution of this paper is a finding that the representer theorem, one of the most desirable mathematical property for machine learning problems, holds for the augmented permanental process. The finding brings a modern SOTA algorithm to the practically important problem "inferring point process intensity as function of covariates", which has been overlooked in the recent machine learning community. Although the key derivation of algorithms are based on the methodology proposed in [Hideaki Kim 2021], we believe that the result makes a non-incremental impact on the Bayesian inference community for point processes. Also, our finding makes a non-trivial contribution to the reproducing kernel Hilbert space community because it shows a novel and practically important likelihood function (Eq. (2)) for which the representer theorem holds under the regularization term of the squared Hilbert space norm.
>
> We will add the explanation of our contribution in the main text. We hope that our explanation is now satisfactory to the reviewer.
>
>
>
> **Can you provide the running time comparisons of your method against the conventional Bayesian method, such as INLA, for the final experiments?**
>
> We apologize for the confusing notations, but we showed the running times (in seconds) of all methods in Table 1 and 2, which are denoted by $\tau(s)$. For example in Table 2 copper, the average running times of INLA and APP$_{\text{RFM}}$ are 82.9 sec and 2.12 sec, respectively. We will add the explanation of notation in Table 2 legend. Also, we performed a more detailed comparison of running times based on a larger synthetic data (see our 2nd response to reviewer Gqug).
>
>
>
> **I expect the authors to present more in-depth analysis about the limitations of the method.**
>
> According to the reviewer's suggestion, we will add detailed discussions about the possible limitations of our method as follows.
>
> This time, we optimized the hyperparameter of our model by maximizing the marginal likelihood through grid search, but it is clearly beneficial if gradient descent algorithms could be utilized. Actually, the maximization of the marginal likelihood, which has an argmin (see Section 2.2 and Eq. (10)), belongs to a bi-level optimization, and we can compute gradients of the marginal likelihood regarding the hyperparameter exactly  [a] . Better hyperparameters could be found by implementing the gradient descent methods.
>
> We achieved a scalable algorithm under a degenerate form of equivalent kernel function, but an equivalent kernel function with a small rank might lead to overconfident predictive covariance [b]. A promising approach to the problem is employing variational Bayes (VB) approximations with inducing points, which usually provide a full-rank predictive covariance.  VB-based method for the scenario "inferring intensity as function of covariates" have not been proposed so far, and thus it is an important next step of our study.
>
> As in ordinary permanental processes, the nodal line problem could arise in the augmented permanental process (APP): the posterior distribution of the latent variable $x(\cdot)$ has many local modes since $\pm x(\cdot)$ can lead to similar intensity functions $\lambda = x^2(\cdot)$, and artificial zero crossings of $x(\cdot)$ could happen especially on a location where the intensity is row. To alleviate the problem, John and Hensman have proposed to extend the quadratic link function to include an offset parameter $\beta$, so that $k(\cdot) = (x+\beta)^2$ (see [23] in References), which is also valid for our APP.
>
> [a] Stephen Gould et al.. On differentiating parameterized argmin and argmax problems with application to bi-level optimization. arXiv preprint arXiv:1607.05447, 2016.
>
> [b] Carl Edward Rasmussen and Joaquin Quinonero-Candela. Healing the relevance vector machine by
> augmentation. In International Conference on Machine Learning, 2005.

---

> > ### Comment · Area_Chair_Vp4P · 2022-08-08
> > **re: rebuttal**
> >
> > Hi reviewer oHGh, thanks for your review. Did the author's rebuttal satisfy your curiosity and are you standing by your score?

---

> > ### Comment · Reviewer_oHGh · 2022-08-09
> > **Feedbacks**
> >
> > Thanks for your clarifications! I changed my score to marginally above based on your rebuttal.

---

### Official Review · Reviewer_Gqug · 2022-07-07

**Rating:** 5
**Confidence:** 2
**Soundness:** 3 good
**Presentation:** 2 fair
**Contribution:** 2 fair

**Summary:**

The paper proposes a new inference method for a variant of a Gaussian Cox process, which the authors call augmented permanental process (APP). The model is described as a point process that uses a Gaussian process on the covariate space with an intensity function based on a square link function. The main focus of the paper is to develop a tractable Bayesian inference to estimate the posterior distribution of the latent intensity function that is based on a novel application of the represent theorem. In the next step the predictive distribution is approximated using a Laplace approximation of the posterior. The method is evaluated on a synthetic and three real-world datasets.

Update after rebuttal: increased score to 5.

**Questions:**

Could you provide a discussion of the accuracy of your approximation methods? How much does the Laplace approximation affect the results? What is the effect of approximating the kernel?

For me the discussion of the "degenerate approach" came a bit late. In the section before it was always given that we have degenerate kernel. It would be helpful to emphasize in the beginning that we can obtain it by different approximation techniques.

Here you could also discuss the connection to inducing point GPs (which are related to the Nystrom approximation)?

**Limitations:**

The authors discussed the limitations.

**Strengths And Weaknesses:**

*Strengths:*
The paper is well written and the math seems to be sound.

The paper develops interesting theory deriving a representer theorem for the latent intensity function. Using this connection the authors obtain a MAP estimator that scales linearly in the number of datapoint. The theoretical insights could be used beyond the scope of the paper.

*Weaknesses:*
The empirical evaluation of the method is not overwhelming.

First, on all four considered datasets the benefits of using the new method seem to be rather marginal. The performance gaps to the reference methods are rather small. For instance, when comparing KIE_ra_SI and APP_NYS on the real-world datasets we get the log-likelihood values  5.56 vs 5.56, 2.92 vs. 2.87, and 2.39 vs. 2.36. It is not clear to me if these improvement are of any practical value. If the authors belief so, could you find another metric or experiment demonstrating the benefit. If the benefit is speed (at least in comparison to INLA), it would be helpful demonstrating the benefit clearer. For instance, you could show a plot with dataset size on the x-axis and inference speed/prediction performance on the y-axis.

Second, the datasets seem rather small. It would be interesting to see how those algorithms scale with increased dataset size (see above). Is this an area where you method can shine? For instance, due to the linear scaling can we use APP on huge datasets where competing methods fail?

As stated above the paper makes interesting theoretical contributions which is focused on the proposed model. If not demonstrating clear empirical benefits, another way where the paper could have impact would be to extend the theory and to show how similar ideas could be applied beyond the scope of the specific model.

The presentation is quite dense and might be improved by some visual guiding, e.g., a model diagram.

---

> ### Author Response · Authors · 2022-08-02
> **Responses to Reviewer Gqug**
>
> We would like to thank the reviewer for pointing out our insufficient explanations and providing important perspectives to discuss in the paper. Below we provide a detailed response to each of the major and minor comments, which we believe is satisfactory.
>
>
>
> **First, on all four considered datasets the benefits of using the new method seem to be rather marginal. The performance gaps to the reference methods are rather small. It is not clear to me if these improvement are of any practical value.**
>
> We will add in the paper the following discussion about the experimental results, which we hope makes the benefit of our method clear.
>
> In the experiments on the six synthetic data sets, APP achieved significantly better predictive performances than the reference methods across all the data sets, where the performance gaps were not marginal (e.g., the improvements of the integrated absolute error $l_\{.5\}$ were 9\% ~ 45\%). For example, given an intensity function $\lambda(t)$, the predictive mean of event count observed in a region $\mathcal{T}$ is $\int_{\mathcal{T}} \lambda(t) dt$, and thus the improvement of intensity estimation directly affects the accuracy of event count prediction, which is a practically important task.
>
>  In the experiments on the real-world data "bei", APP achieved the significant 10\%~20\% improvements of $cl_\{\text{test}\}$ against the reference methods, which we believe were not marginal. On "copper", the predictive performances of all the methods were found not to be significantly different from each other, which might be due to that the data size was so small ($N = 67$). On "clmfires", the APP's improvements of predictive performance against INLA were small, but being statistically significant for $ll_\{\text{test}\}$.
>
> We confirmed that our APP achieved significantly substantial improvements of predictive performances on many of the data sets, while on some data sets the improvements were significant but small.
>
> **If the benefit is speed (at least in comparison to INLA), it would be helpful demonstrating the benefit clearer. For instance, you could show a plot with dataset size on the x-axis and inference speed/prediction performance on the y-axis.**
>
> According to the reviewer's suggestion, we ran an additional experiment based on larger data sets, and performed a comparative study of INLA and APP. Please see Section S6 in the revised supplementary material. Note that we here show the results in tabular form, but we promise to add in the paper a plot with dataset size on the x-axis and inference speed/prediction performance on the y-axis.
>
>
> **Could you provide a discussion of the accuracy of your approximation methods? How much does  the Laplace approximation affect the results? What is the effect of approximating the kernel?**
>
> It is difficult to analyze the accuracy of the Laplace approximation analytically. Rather, we confirmed in the experiments that the predictive distribution and the marginal likelihood obtained by using the Laplace approximation worked well. Table 1 shows that our methods (APP$_*$) achieved better predictive performances than the reference methods for various quantile losses ($l_\{.025\}$, $l_\{.5\}$, $l_\{.975\}$), which suggests that the Laplace approximation can provide a reasonable approximation of the predictive distribution. Table 1 also shows that our model optimized with the approximate marginal likelihood (APP$_\{\text{RFM}\}$) outperformed our model with an initial hyperparameter (APP'$_\{\text{RFM}\}$), suggesting empirically that the Laplace approximation can provide a reasonable approximation of the marginal likelihood. Although these discussions are mentioned briefly in Section 4.1, we will revise the manuscript by adding a more detailed discussion.
>
> As for the effect of approximating the equivalent kernel function, see our 1st response to reviewer bJU1 and our 3rd response to reviewer oHGh.
>
> **For me the discussion of the "degenerate approach" came a bit late.**
>
> According to the reviewer's suggestion, we will move Section 2.4 "Construction of Equivalent Kernel" to the next to Section 2.2 "Maximum a Posteriori Estimator".
>
> **Here you could also discuss the connection to inducing point GPs (which are related to the Nystrom approximation)?**
>
> Thank you for the interesting suggestion. Some of inducing point GPs (e.g. Subset of Regressors algorithm) approximate a kernel function by a degenerate form of function, by which we can construct a degenerate equivalent kernel. Inducing point GP and Nystr\"{o}m approximation are attractive in that they are applicable to arbitrary kernel functions, while RFM approximation is limited to additive and shift invariant kernels. See also the discussion about variational Bayes approaches for inducing point GPs (our 3rd response to reviewer oHGh). We will add the discussion in the main text.

---

> > ### Comment · Reviewer_Gqug · 2022-08-08
> > **Reply**
> >
> > Thank you for your response. I appreciate your defense of the log-likelihood improvements but I still can't see that the performance gaps are very convincing. Again, if you could show that your method also achieves a great speed up it would make that paper stronger.
> >
> > > Re: Speed vs. performance plot.
> >
> > Thanks for adding the table. Unfortunately, it is very hard to see what's going on just looking at this large table. I plot would have been much more helpful, and I can't really judge the speed benefits from this plot (even though you promised to make it later). And don't see why a plot would have been more difficult to make than a table?
> >
> > In the light of the other reviews and the weaknesses I pointed out, I, unfortunately, won't recommend acceptance. However, I think the paper has potential and encourage the authors to keep working on it and to make the selling points more convincing.

---

> > > ### Author Response · Authors · 2022-08-09
> > > **2nd response to Reviewer Gqug**
> > >
> > > We would like to thank the reviewer for kindly reading our responses!
> > >
> > > > Unfortunately, it is very hard to see what's going on just looking at this large table. I plot would have been much more helpful,
> > >
> > > Sorry for the inconvenience. Actually, we were not familiar with the OpenReview system, and failed to upload the manuscript with a plot before Aug 03...
> > >
> > > As a response to the reviewer's 2nd-round comment, we made two figures about speed vs. performance in the supplementary materials! Figure S1 displays the predictive performances and the CPU times as function of the data size, which shows that our APP$_\{\text{RFM}\}$ achieved better predictive performances than INLAs, while APP$_\{\text{RFM}\}$ was performed substantially faster than INLAs for at least up to tens of thousands of data points. Figure S2 displays the predictive performances multiplied by the CPU times (the lower, the better), which represent the predictive performances penalized by its execution times, that is, the ratios of predictive performance to speed. Figure S2 also shows that our APP$_\{\text{RFM}\}$ is beneficial against INLAs.
> > >
> > > We hope that the figures would alleviate the reviewer's concerns.

---

> > > > ### Comment · Reviewer_Gqug · 2022-08-09
> > > > **Thanks**
> > > >
> > > > Thanks for updating the figure. I stand by the point that the results are promising but not clearly impactful. Still I appreciate the effort and I will increase my score to 5.

---

### Official Review · Reviewer_bJU1 · 2022-07-11

**Rating:** 7
**Confidence:** 4
**Soundness:** 3 good
**Presentation:** 3 good
**Contribution:** 3 good

**Summary:**

This paper proposes an efficient Beyian algorithm to estimate the intensity function as a function of Covariates in a Reproducing Hilbert Kernel Space (RHKS). The purposed algorithm is based on an augmented permanent process and enjoys significant computational advantages. The effectiveness of the proposed algorithm is demonstrated through numerical experiments.

**Questions:**

The proposed approach is essentially an extension of Flaxman et al (2017), which focuses on estimating intensity function as a function of time/spatial locations. But I think the extension is meaningful and solves an important problem. My only concern is about the computation of the Equivalent Kernel in Section 2.4. Because of the change of support from t to y, the integral equation (9) indeed can be computationally intensive to solve when the dimension of covariates is high. This somehow offsets some benefits of kernel ridge regression in RHKS, which can be used to fit high-dimensional functions using kernels such as the multiplicative Gaussian kernel. The algorithm in 2.4 essentially proposes to approximate the integral equation (9) through J-point numerical integration. However, when the dimension of y is large, the required J would grow exponentially, and the proposed approach cannot be applied to the multiplicative Gaussian kernel. One possible solution is perhaps to use a stochastic approximation of (9) in a similar fashion to those in  [1]. It will be helpful to give some discussion on this issue.

In addition, an important class of nonparametric models in Statistics is the additive model. So if the reproducing kernel K(.,.) is additive, would the Equivalent Kernel also be additive? If this is the case, the computation of the equivalent kernel can be greatly reduced for the additive model, which would be a great contribution as well.

The simulation study only covers 1-D functions. It would be better if at least 2-D functions can be studied, just to be consistent with the real data analysis.

Finally, the title seems to emphasize "inference" using the proposed method. But I do not see any inferential results in numerical studies. What kind of inferences can be carried? Can you give some examples?

[1] Xu, G., Waagepetersen, R. and Guan, Y., 2019. Stochastic quasi-likelihood for case-control point pattern data. Journal of the American Statistical Association, 114(526), pp.631-644.

**Strengths And Weaknesses:**

Strengths: the paper is well written, the presentation is clear,  the motivations are valid and the proposed approach is novel and effective.
Weaknesses: there might still be computational bottlenecks when the number of covariates is large; the synthetic data experiment can be improved.

---

> ### Author Response · Authors · 2022-08-02
> **Responses to Reviewer bJU1**
>
> We would like to thank the reviewer for the highly positive comments and suggestions, by which we are strongly encouraged. Below we provide a detailed response to each of the comments.
>
>
>
> **My only concern is about the computation of the Equivalent Kernel in Section 2.4. ...  The algorithm in 2.4 essentially proposes to approximate the integral equation (9) through  J-point numerical integration. However, when the dimension of y is large, the required J would grow exponentially, and the proposed approach cannot be applied to the multiplicative Gaussian kernel.**
>
> Thank you for casting an important point of view. In fact, the required number of integration points, $J$, does not grow with the dimension of $\mathcal{Y}$ (covariate domain), but does with the dimension of $\mathcal{T}$ (observation domain). It is easily verified by the fact that the integral operator over $\mathcal{Y}$ in Eq. (9) can be transformed into the operator over $\mathcal{T}$ (see Eq.(4)), and that the $J$ integration points are selected not on  $\mathcal{Y}$ but on $\mathcal{T}$ (see Section 2.4.1). Point processes are usually utilized to analyze at most three-dimensional data ($\mathcal{T} \subset \mathbb{R}^{3}$), and thus the required $J$ practically stays within a feasible size regardless of the dimension of $\mathcal{Y}$. An intuitive explanation for the requirement of $J$ is that observed covariate $y(t)$ exists on a lower-dimensional manifold ($\rho(y)$ has the information about the manifold) in high-dimensional domain $\mathcal{Y}$, and the effective volume of integral region is rather small.
>
> The discussion so far is about the approximation error of the integral operator. Next, we discuss about the approximation error of the equivalent kernel under the approximate integral. Let the integral operator in Eq. (9) and its approximation through J-point numerical integration be denoted by $\mathcal{K}$ and $\mathcal{K}_J$, respectively. Then the equivalent kernel to solve Eq. (9) has the relation (for detailed derivation, see [3] in References),
> $$
> h-h_J=(\mathcal{I}+2h_J)^{-1}(\mathcal{K} h-\mathcal{K}_J h),
> $$
> where $h$ and $h_J$ are the solutions of Eq. (9) under $\mathcal{K}$ and $\mathcal{K}_J$, respectively. The relation shows that the speed of convergence of $h_J$ to $h$ is at least as rapid as the speed of convergence of the integral operator, $\mathcal{K}_J h$ to $\mathcal{K} h$. Therefore, the required $J$ to obtain a good approximation of the equivalent kernel also stays within a feasible size, as long as the dimension of $\mathcal{T}$ is low.
>
> We will add the discussion in the paper, which highlights the merits of our proposed model.
>
>
>
> **One possible solution is perhaps to use a stochastic approximation of (9) in a similar fashion to those in  [1]**
>
> Thank you for the valuable suggestion. Unfortunately, we cannot obtain the paper before the end of the author response period, but we promise to add a discussion about the reference, if accepted.
>
>
>
> **if the reproducing kernel K(.,.) is additive, would the Equivalent Kernel also be additive?**
>
> Thank you for the interesting suggestion, but unfortunately, the equivalent kernel is not additive even if the  reproducing kernel is additive.
>
>
>
> **The simulation study only covers 1-D functions. It would be better if at least 2-D functions can be studied**
>
> Unfortunately, we cannot finish a simulation study of 2-D covariate before the end of the author response period. We plan to create a synthetic data set ($\mathcal{T}$ $\subset$ $\mathbb{R}^{2}$, $\mathcal{Y}$ $\subset$ $\mathbb{R}^{2}$) generated from the following intensity function: $\lambda(t) = g_1(d_R(t))+g_1(d_Z(t))$, where $d_R(t)$ and $d_Z(t)$ denote the shortest distances from a given location $t$ to the set of lines arranged in the shapes of the letter R and Z, respectively.
>
>
>
> **the title  seems to emphasize "inference" using the proposed method. But I do not see any inferential results in numerical studies. What kind of inferences can be carried?**
>
> We assume that the word "inference" includes the estimation of intensity function and the prediction of event locations/counts in a given region. In point processes, these "inference" tasks can be performed by using the predictive distribution of each target, of which accuracy we evaluated in Experiments based on the $\rho$-quantile loss in Table1 (estimation of intensity), the negative test log likelihood of point patterns (prediction of event locations) and the negative test log likelihood of counts in Table 2 (prediction of event counts). Details of the metrics are given in the supplementary material (S5.3). We will add the explanation about the inference tasks that can be performed in point process models.

---

> > ### Comment · Area_Chair_Vp4P · 2022-08-08
> > **main point**
> >
> > Hi reviewer bJU1, did the authors satisfy your curiosity on your main point?

---

> > > ### Comment · Reviewer_bJU1 · 2022-08-09
> > > **response to main point**
> > >
> > > Yes, the clarification looks good to me.

---

> > ### Comment · Reviewer_bJU1 · 2022-08-09
> > **Response to response**
> >
> > Thank you for the clarification. Just one minor issue.
> >
> > In Statistics literature, the term "Inference" typically refers to procedures that quantify the uncertainty in estimation, such as confidence intervals, prediction intervals, and hypothesis testing. You may consider changing the word "Inference" to "Modelling" to avoid confusion. But I will not insist on it.

---

> > > ### Author Response · Authors · 2022-08-09
> > > **Re: Response to response**
> > >
> > > We would like to thank the reviewer for the informative comment. According to the reviewer's suggestion, we will change the word "inference" to "modelling". In this context, we would appreciate the reviewer if the reviewer gives a comment about whether the word "estimation" can avoid the confusion caused by "inference" or not.
> > >
> > > We also really would like to thank the reviewer for the helpful comment about our discussion with Reviewer V1pm!

---

### Official Review · Reviewer_V1pm · 2022-07-14

**Rating:** 3
**Confidence:** 4
**Soundness:** 2 fair
**Presentation:** 2 fair
**Contribution:** 2 fair

**Summary:**

This paper proposes an approach for making Maximum A Posteriori (MAP) inference on the intensity function of a Poisson process, regarded as a function of covariates. The representer theorem, which the authors show applies when a Gaussian Process prior is placed on the square root of the intensity function, is used to achieve a complexity linear in the sample size.

**Questions:**


Why do we need Augmented Permanental Processes? Why not use Permanental Processes in the covariates space?

**Strengths And Weaknesses:**


**It is unclear why this work is needed. Do we really need Augmented Permanental Processes?**

There is nothing in the theory of Poisson processes that requires the input space to be time, space or space-time.

If the authors want to infer the intensity function as a function of covariates, they could simply define the space of points to be the space of covariates, namely $\mathcal{Y}$.

The Poisson likelihood would simply read $p(\mathcal{D} \vert  x(\textbf{y}))= \sum_{i=1}^n \log \lambda(\textbf{y}_i) - \int_\mathcal{Y} \lambda(\textbf{y}) d\textbf{y}, ~~~ \lambda(\textbf{y}) := \kappa(f(\textbf{y})),$ and the prior would read $f \sim \mathcal{GP}(0, k).$

For all practical purposes, this would work even if the mapping function $g: \textbf{t} \to \textbf{y}$ is not injective.

Which begs the question: what contribution does this work make to the literature on Permanental Processes [14, 24, 46]?

---

> ### Author Response · Authors · 2022-08-02
> **Responses to Reviewer V1pm**
>
> We would like to thank the reviewer for the comment, to which we provide a detailed response below. We believe that we can fully dispel the critical concern about the significance of our paper, which arises from the reviewer's unfortunate misunderstanding. We apologize for our confusing explanation.
>
>
>
> **If the authors want  to infer the intensity function as a function of covariates, they could simply define the space of points to be the space of covariates. ... It is unclear why this work is needed. Do we really need Augmented Permanental Processes?**
>
> Yes, we do. Ordinary permanental processes defined in the covariate space cannot address the problem considered in our paper, and thus the augmentation of the processes is necessary.
>
> We consider the situation that event points are generated on a domain $\mathcal{T}$ based on a Poisson intensity $\lambda (t \in \mathcal{T})$, and that the intensity value $\lambda (t)$ at each point in $\mathcal{T}$ is determined by the covariate $y(t)$ at the point, where the intensity is represented by using a function of covariates, $f(y)$, as $\lambda(t) = f(y(t))$. It is totally different from the situation, suggested by the reviewer, that event points are generated on the covariate domain $\mathcal{Y}$ based on a Poisson intensity $\lambda (y \in \mathcal{Y})$. Of course, $f(y)$ is different from $\lambda (y)$. The difference becomes clearer by comparing the likelihood functions for the augmented permanental process,
> $$
> \log p(\mathcal{D}|x(y)) = \sum_{n=1}^N \log  f(y_n) - \int_\mathcal{T}  f(y(t))dt, \quad y_n = y(t_n),
> $$
>  with that for the ordinary permanental process suggested by the reviewer,
> $$
> \log p(\mathcal{D}|x(y)) = \sum_{n=1}^N \log  \lambda(y_n) - \int_\mathcal{Y}  \lambda(y) dy.
> $$
> As we show in Eq. (4), the former survival term, $\int_\mathcal{T} f(y(t))dt$, can be transformed into $\int_{\mathcal{Y}} \rho(y)f(y)dy$, which is inconsistent with the latter survival term, $\int_{\mathcal{Y}} f(y)dy$, since $\rho(y) \neq 1$ in general. As a result, the MAP estimator for the former likelihood, $f(y)$, differs from the MAP estimator for the latter one, $\lambda(y)$, indicating that ordinary permanental processes simply defined in the covariate space cannot address the problem considered in our paper. The discussion stated above holds true regardless of the dimension number and the physical meaning (time, space or space-time) of $\mathcal{T}$.
>
> We recognize from the reviewer's comment that the problem we tackled is a little confusing. Thus, we promise to improve the explanation of the problem we tackled by including the above discussion in the main text. We hope that the reviewer's concern is now fully cleared.

---

> > ### Comment · Reviewer_V1pm · 2022-08-06
> > **Re: Responses to Reviewer V1pm**
> >
> > Thanks for your response, but you seem to have missed the essence of my question, so I'll give it another go.
> >
> > **Question:**
> >
> > Fundamentally, is there ever a case in practice where the log-likelihood model $$\log p(\mathcal{D} | x(y)) = \sum_{n=1}^N \log \lambda(y_n) - \int_{\mathcal{T}} \lambda(y(t)) dt$$ (i.e. the Augmented Permanental Process log-likelihood; APP) makes sense but no log-likelihood model of the form $$\log p(\mathcal{D} | x(y)) = \sum_{n=1}^N \log \lambda(y_n) - \int_{\mathcal{Y}} \lambda(y) dy$$ (i.e. the Permanental Process and traditional point process log-likelihood; PP) makes sense or as much sense as the former log-likelihood?
> >
> > **Answer:** No.
> >
> > **Corollary:** There is no case in practice where the Augmental Permanental Process is needed.
> >
> > **Reasoning:** Let's assume points are collected as $(t_i)$ from which we may generate covariates $(y_i=y(t_i))$ using the feature/covariate function $y$.
> >
> > Consider two operators A and B tasked with learning the intensity function as a function of covariates: $\lambda : \mathcal{Y} \to \mathbb{R}^+$.
> >
> > A is given samples $(y_1, \dots, y_N)$. B is given samples $(y_1, \dots, y_N)$, but also told that they were generated from samples $(t_1, \dots, t_N)$ using covariate function $f$.
> >
> > Claiming that the APP likelihood can possibly make more sense than the PP likelihood is equivalent to saying that, when modeling point processes, it can help to know how the inputs were generated...
> >
> > This clearly makes little sense.
> >
> > **Concrete Illustration:** Imagine $(y_n)$ are GPS coordinates, and you are tasked with modeling a spatial point process in the GPS coordinates space. Would it help you in any way to know that those GPS coordinates were in fact generated from geodetic coordinates, and be given the mapping from geodetic to GPS coordinates? What if GPS coordinates were generated directly, would it make a difference?

---

> > > ### Author Response · Authors · 2022-08-08
> > > **Re: Re: Responses to Reviewer V1pm**
> > >
> > > Thank you for the response. We think that the disagreement between the reviewer and us about the situation/problem tackled in the paper could be relieved by discussing "Concrete Illustration" the reviewer provided.
> > >
> > > In Concrete Illustration, the reviewer seems to consider the situation that event points ($t_n$) are observed on geodetic coordinates ($\mathcal{T}$), and that covariates are given as GPS coordinates ($y_n=f(t_n) \in \mathcal{Y}$) (Is it correct?). Then the reviewer concludes that it makes no sense to consider whether the point process is defined on GPS coordinates or geodetic ones, because it is just a change of coordinate. If our understanding is correct, the situation the reviewer considers here is NOT the situation considered in our paper: In our situation, covariates are generally not coordinates but spatial covariates (e.g., slope and elevation), which are given or observed at every point on geodetic (or equivalently, GPS) coordinates as $f(t)$. The "spatial" covariates differ from the "GPS coordinate" covariates essentially in that the "spatial" covariates can have the same values at different points of geodetic coordinates ($t \in \mathcal{T}$), which often happens in practice (see Figure 2A), while the "GPS coordinate" covariates should have different values at different points of geodetic coordinates. In other word, the "spatial" covariate map $f(t)$ can be non-bijective, but the "GPS coordinate" covariates should be bijective. The reviewer's claim is correct only if the covariate map is bijective (it is just a change of coordinate, as mentioned in Section 2.2), but we consider the non-bijective covariate map in the paper.
> > >
> > > Next, we explain why the augmented permanental process is necessary for the situation we consider, and why ordinary permanental process simply defined in the spatial covariate space is not appropriate to our situation. To make it clearer, consider a simple situation that the two-dimensional geodetic (or equivalently, GPS) domain $\mathcal{T}$ is partitioned into 2 $\times$ 2 regular sub-areas, and that the intensity is constant over each sub-area. Let the index of each sub-area be denoted by $(i,j)$ for $ 1\leq i, j \leq 2$, and we assume that the observed spatial covariate values $y_\{i,j\}$ at the four sub-areas are $y_\{1,1\} =  y_\{1,2\} = y_\{2,1\} = a$, and $y_\{2,2\}=b (a\neq b)$. Then, it is clear that the intensity values for covariate $y=a$ and covariate $y=b$ should be estimated as $\lambda_\{y=a\} = (N_\{1,1\}+N_\{1,2\}+N_\{2,1\})/(3S)$ and $\lambda_\{y=b\} = N_\{2,2\}/S$, respectively, where $N_\{i,j\}$ and $S$ are the number of observed events at sub-area $(i,j)$ and the area of each sub-area, respectively. Here, we can evaluate the intensity value at each sub-area of $\mathcal{T}$ as $\lambda_\{1,1\} = \lambda_\{1,2\} = \lambda_\{2,1\} = \lambda_\{y=a\} = (N_\{1,1\}+N_\{1,2\}+N_\{2,1\})/(3S)$, and $\lambda_\{2,2\} = \lambda_\{y=b\} = N_\{2,2\}/S$. Summing up all the sub-areas at which the covariate has the same value  ($S+S+S=3S$ for $\\{(1,1), (1,2), (2,1)\\}$) is essential task in our augmented permanental process, in which the weight distribution of the covariate $\rho(y)$ in Eq. (3) represents the effective size of the sum of sub-areas at which covariate value $y$ is observed. However, if we employ the ordinary permanental process simply defined in the spatial covariate space, then $(N_\{1,1\}+N_\{1,2\}+N_\{2,1\})$ event points are concentrated on the point $y=a$, resulting in the intensity estimation of an excessive value at $y=a$. Therefore, the augmented permanental process is necessary.
> > >
> > > We hope that our explanation is now satisfactory.

---

> > > > ### Comment · Reviewer_V1pm · 2022-08-08
> > > > **Re: Re: Re: Responses to Reviewer V1pm**
> > > >
> > > > **RE: MISUNDERSTANDINGS**
> > > >
> > > > I appreciate the effort to further clarify possible misunderstandings.
> > > >
> > > > However, kindly note that the situation described in my previous response is exactly the same situation your paper deals with, and which you made even clearer in the response above.
> > > >
> > > > The only apparent difference, which you seem to be heavily relying on, is that in your example, the covariate function $y: \mathcal{T} \to \mathcal{Y}$ is not injective.
> > > >
> > > > Note that while the mapping from geodesic to GPS in my example was injective, this was not a requirement for my argument to hold! As mentioned in my very first review, the fact that $y$ is not injective poses no problem, theoretical or practical, to learning the intensity function directly in the covariate space from covariate-space observations $(y_1, \dots, y_n)$. It certainly does not pose any more problems than the APP you are proposing...
> > > >
> > > > **RE: YOUR EXAMPLE**
> > > >
> > > > I have to say I'm having a hard time making sense of your example...
> > > >
> > > > Why would the intensity estimation at $y=a$ be 'excessive' in the PP case but not in the APP case?!
> > > >
> > > > In both cases the number of points concentrated at $y=a$ in the covariate space is the same... In both cases the intensity function should be estimated the same...
> > > >
> > > > But what's really troubling is that you would model this problem in the covariate space through an intensity function to start with. This 'function' in the covariate space is literally defined on a set with two elements $(a, b)$.
> > > >
> > > > You can use the scale-density/pmf formalism of point processes to model these points in the covariate space: $$n=n_a+n_b \sim Poisson(\lambda),$$ $$\lambda= \lambda_a + \lambda_b,$$ $$p_a = \frac{\lambda_a}{\lambda},$$ and you consider that out of $n$ i.i.d. draws from a Bernoulli taking values $(a, b)$ with probabilities $(p_a, 1-p_a)$, we observed $n_a$ times the value $a$.
> > > >
> > > > This is a classic Bayesian **parametric** problem that admits a conjugate solution.
> > > >
> > > > Expressing the intensity function in terms of $t$ is also straightforward.
> > > >
> > > > There is no need for Augmented Permanental Processes here. On the contrary, by generating the samples $(y_1, \dots, y_n)$, you can see that points in the covariate space are concentrated at two locations, which should make you reconsider the need for an intensity function in the covariate space.
> > > >
> > > > **RE: THE FUNDAMENTAL ISSUE REMAINS**
> > > >
> > > > Every time you might be tempted to model a phenomenon $(t_1, \dots, t_n)$ as an APP in the covariate space with covariate mapping function $y$, you ALWAYS HAVE THE ALTERNATIVE to consider modeling the points $(y_1, \dots, y_n)$ (where $y_i = y(t_i)$ directly in the covariate space as a point process, for instance a permanental point process.
> > > >
> > > > I see nothing in your paper or your response that justifies *why not*.
> > > >
> > > > In fact PP in the covariate space is the closest alternative to your proposed model, and you should have included it in your experiments to support any claim that it would have underperformed APP.

---

> > > > > ### Comment · Reviewer_bJU1 · 2022-08-09
> > > > > **My two cents**
> > > > >
> > > > > I think the issue here is that the target of the estimation is the intensity, $\lambda(t)$, defined on the spatial/temporal domain $(t_1,\cdots,t_n)$. If one switches to the covariate space $y_1,\cdots,y_n$, the definition of the intensity changes from $\lambda(t)$ to $\lambda^*(y)$. These two intensities may have completely different values and different physical interpretations. For example, consider the occurrence of criminal activities in a city, and let's say we have a covariate (medium household income, y). People are typically interested in the crime intensity of a particular location $t$, i.e., ($\lambda(t)$), and its relation to the local household income ($y(t)$), which can be estimated using the proposed model. However, it would make little practical sense to estimate the crime intensity in the covariate space, i.e., $\lambda^*(y)$, which has a very different physical interpretation (what is the crime intensity for household income range 10k-11k??). The physical interpretations may be even harder when the dimension of the covariate space is high. Therefore, in my opinion, it still makes sense to use APP for many applications in spatial/temporal point processes, simply because the physical interpretations of the intensity in space/time are more relevant.

---

> > > > > > ### Comment · Reviewer_V1pm · 2022-08-09
> > > > > > **Re: My two cents**
> > > > > >
> > > > > > I think you are contributing to making my point in that you seem to be thinking that point processes and their intensities only make sense on spatio-temporal domains, which is not true.
> > > > > >
> > > > > > For instance, in your example the intensity function defined in the covariate/income space does have as much practical and intuitive meaning as the intensity function defined as a function of space.
> > > > > >
> > > > > > Namely, the integral of the intensity function (regarded as a function of income) over an interval, say [10k, 11k], gives you the expected number of criminal activities taking place in houses with median household income between 10k and 11k. The same way that the integral of the intensity function (regarded as a function of space) over an neighborhood gives you the expected number of criminal activities taking place in the neighborhood.
> > > > > >
> > > > > > This practical interpretation does not change as a function of the dimension of the input/feature space. More importantly, the proposed model is primarily affected by dimension of the feature space, and does not scale any better than modeling the process in the feature/covariate space directly. In fact, for all practical purposes the proposed model is identical to applying the Permanental Process in the feature space, except for the added weight term $\rho(y)$ inside the integral in the Poisson likelihood.
> > > > > >
> > > > > > Taking a step back, in your example, if we are given crime locations $(t_1, \dots, t_n)$ and the associated median household incomes at crime locations $(x_1, \dots, x_n)$, I can think of 3 possible scenarios/needs:
> > > > > >
> > > > > > - **Scenario 1:** We want to study the effect of location on crime. In this case we can model $(t_1, \dots, t_n)$ as a point process, and perform standard Bayesian nonparametric inference on its intensity function (e.g. using a permanental process prior).
> > > > > > - **Scenario 2:** We want to study the effect of household income on the occurrence of crime in the household. In this case we can model $(y_1, \dots, y_n)$ as a point process; there is nothing odd about this. Even if the set of possible values for $y$ is best modelled as a small finite set, we could still model this point process very easily and efficiently using standard Bayesian parametric methods. For instance we could place a Poisson distribution prior on the total count, and an independent Dirichlet distribution prior on distribution of $y$ values among possible values.
> > > > > > - **Scenario 3:** We want to study the joint effect of location and income on crime. In this case we can model $(z_1, \dots, z_n)$ as a point process where $z_i = (t_i, y_i)$. Again, we could perform  standard Bayesian nonparametric inference on its intensity function (e.g. using a permanental process prior).
> > > > > >
> > > > > > I do not see why or when we would need APPs. The case where the covariate mapping function $y$ is not injective falls in scenario 3. E.g. each house has a single median household income, but two houses can share the same median income.

---

> > > > > > > ### Comment · Reviewer_bJU1 · 2022-08-09
> > > > > > > **My two cents-continued**
> > > > > > >
> > > > > > > I am not saying that "point processes and their intensities only make sense on spatio-temporal domains". All I am saying is that in many applications, the primary research interest is to model the intensity of point processes in spatio-temporal domain. I would even say that this is the primary purpose for most point process applications. If the primary purpose is to estimate the spatial/temporal intensity, your scenarios 2-3 do not apply. The proposal in scenario 1 is too simplistic. In many applications, researchers are generally more interested in studying what covariates cause the elevation/decrease of intensities in a location $t$.

---

> > > > > ### Author Response · Authors · 2022-08-09
> > > > > **Re: Re: Re: Re: Responses to Reviewer V1pm**
> > > > >
> > > > > We really appreciate the reviewer's patient discussion with us. As Reviewer bJU1 kindly gave a comment, the two intensities, for APP and ordinary permanental processes defined in the covariate space, may have completely different values and different physical interpretations. We believe the clear significance of APP, and we sincerely would like to dispel the unfortunate disagreement. Because a further detailed response to the reviewer's honest comments might be a burden for both the reviewer and us, we give a brief comment to the following reviewer comment, which we believe is a critical point for our discussions so far.
> > > > >
> > > > > > As mentioned in my very first review, the fact that is not injective poses no problem, theoretical or practical, to learning the intensity function directly in the covariate space from covariate-space observations $(y_1, y_2, \dots, y_n)$...
> > > > >
> > > > > In our extreme but essential example that the covariate map has only two values ($y(t)=a$ or $y(t)=b$), the event points on the covariate space, $(y_1, y_2, \dots, y_n)$, are concentrated on the point $y=a$, while the event points on the observation space, $(t_1, t_2, \dots, t_n)$ are different from each other. An ordinary permanental processes simply defined in the covariate space cannot explain the observation theoretically, because two event points are never allowed to occur at the same position in point processes. Or equivalently, the intensity value at $y=a$ would be estimated as infinite, which we mentioned as "excessive" in our previous response. This difficulty does not happen when the covariate map is bijective.
> > > > >
> > > > > In our APP, the difficulty never happens because it is a point process defined in the observation space ($\mathcal{T}$), in which two event points never occur at the same position theoretically. We would like to emphasize here that APP estimates the intensity for point process on observation space, $\lambda (t \in \mathcal{T})$, but assumes that the intensity value $\lambda (t)$ at each point in $\mathcal{T}$ is determined by the covariate $y(t)$ at the point (unit of intensity is the inverse of observation space), which never means that APP estimates the intensity for point process on covariate space (unit of intensity is the inverse of covariate space)!

---

> > > > > > ### Comment · Reviewer_V1pm · 2022-08-09
> > > > > > **Last Comment**
> > > > > >
> > > > > > Though you were not able to convince me of the need for this work, I appreciate the effort you put into this discussion and responses to other reviewers' concerns. I offer the following final remarks.
> > > > > >
> > > > > > **1.** As previously mentioned your toy example is a parametric example, and it doesn't make much sense to model it with an APP or a PP for that matter. Note that, in your APP formalism, writing down the intensity function as $\lambda(y(t))$ where $y(t)$ can only take two values unfortunately hides the fact that this function, even as a function of $t$, depends on only two parameters: $\lambda(a)$ and $\lambda(b)$. No need for any Bayesian nonparametrics.
> > > > > >
> > > > > >
> > > > > > **2.** When $y$ is not injective, and yet its image space $\mathcal{Y}$ isn't a finite set, the fact that you can have two points at the same 'location' does not pose a practical problem. You can 'fix' the theory (to be rigorous) by adding an ever small random jitter to all points. In practice, however, you do not need to do this; simply feed all observations to the Poisson likelihood and it will handle duplicates properly, increasing the intensity function as needed. You'll certainly **not** get infinite intensities. The degeneracy of the Gaussian prior would be handle with traditional inducing point techniques.
> > > > > >
> > > > > > There is a parallel between this situation and density estimation. For any two i.i.d. continuous variables $t_i$ and $t_j$, $\mathbb{P}(t_i=t_j)=0$. Yet, when doing density estimation you can find yourself with two identical samples for many reasons (e.g. representation precision, etc.). This is not a reason not to train your density estimator, or to throw away duplicate points (this could adversely affect the learned density). More importantly, this will certainly not cause infinite densities! An intensity function is nothing but a rescaled density.
> > > > > >
> > > > > >
> > > > > > **3.** Strictly speaking the theory breaks down for APP in your toy example as well. Does the Gaussian prior not become degenerate in this case?

---

### Meta-Review · Area_Chair_Vp4P · 2022-08-23

**Recommendation:** Accept
**Confidence:** Less certain

**Metareview:**

The paper looks a the Augmented Permanental point process as a model of (spatial) point phenomena.

One reviewer was unconvinced that the method was needed at all, which the authors refuted. There was a long exchange but I'm on the side of the authors here - the method is clearly distinct from a point process defined on the covariate space. I'm happy that the reviewer was able to make their point and that the discussion was enabled, and I applaud the authors for their patient responses. I'm disregarding that reviewer's score.

The reviewers suggest that an important and interesting contribution is the representer theorem for squared processes. I also appreciated the authors discussion on the approximation error on the integral operator. These should be highlighted in the manuscript.

There was the occasional confusion from the reviewers on notations: for example, the reviewers failed to spot the the performance of the method was presented in the paper using the $\tau$ column in the tables. Please, double check all the reviewer feedback for clarifications.

Overall, I think that there are a couple of interesting ideas in the paper that people working on Point Process data will be impacted by, and am recommending that this is just above the acceptance threshold.

**Award:**

No

---

### Decision · Program_Chairs · 2022-09-14

Accept